# Contextual Multi-Armed Bandits with Communication Constraints

## Abstract

We consider a remote Contextual Multi-Armed Bandit (CMAB) problem, in which the decision-maker observes the context and the reward, but must communicate the actions to be taken by the agents over a rate-limited communication channel. This can model, for example, a personalized ad placement application, where the content owner observes the individual visitors to its website, and hence has the context information, but must convey the ads that must be shown to each visitor to a separate entity that manages the marketing content. In this Rate-Constrained CMAB (RC-CMAB) problem, the constraint on the communication rate between the decision-maker and the agents imposes a trade-off between the number of bits sent per agent and the acquired average reward. We are particularly interested in the scenario in which the number of agents and the number of possible actions are large, while the communication budget is limited. Consequently, it can be considered as a policy compression problem, where the distortion metric is induced by the learning objectives. We first consider the fundamental information theoretic limits of this problem by letting the number of agents go to infinity, and study the regret that can be achieved. In particular, we identify two distinct rate regions resulting in linear and sub-linear regret behaviour, respectively. Then, we propose a practical coding scheme, and provide numerical results for the achieved regret.

## 1 Introduction

In the last few years, synergies between Machine Learning (ML) and communication networks have attracted a lot of interest in the research community, thanks to the fruitful interplay of the two fields in emerging applications from Internet of things (IoT) to autonomous vehicles and other edge services. In most of these applications, both the generated data and the processing power is distributed across a network of physically distant devices, thus a reliable communication infrastructure is pivotal to run ML algorithms that can leverage the collected distributed knowledge (Park et al., 2019). To this end, a lot of recent works have tried to redesign networks and to efficiently represent information to support distributed ML applications, where the activities of data collection, processing, learning and inference are performed in different geographical locations, and should consider limited communication, memory, or processing resources, as well as addressing privacy issues.

In contrast to the insatiable growth in our desire to gather more data and intelligence, available communication resources (bandwidth and power, in particular) are highly limited, and must be shared among many different devices and applications. This requires the design of highly communication-efficient distributed learning algorithms , particularly for edge applications. Information theory, in particular the rate-distortion theory, have laid the fundamental limits of efficient data compression with the aim to reconstructing the source signal with the highest fidelity (Cover & Thomas, 2006b). However, in the aforementioned applications, the goal is often not to reconstruct the source signal, but to make some inference based on that. This requires *task-oriented compression*, filtering out the unnecessary information for the target application, and thus decreasing the number of bits that have to be transmitted over the communication networks. This approach should target the questions of *what* is the most useful information that has to be sent, and *how* to represent it, in order to meet the application requirements consuming the minimum amount of network resources.

Our goal in this paper is to investigate a theoretically grounded method to efficiently transmit data in a Contextual Multi-Armed Bandit (CMAB) problem, in which the context information is available to

a decision-maker, whereas the actions can be taken by a remote entity, called *controller*, controlling a multitude of agents. We assume that a limited communication link is available between the decision-maker and the controller to communicate at each round the intended actions. The controller must decide on the actions to take based on the message received over the channel, while the decision-maker observes the rewards at each round, and updates its policy accordingly.

This scenario can model, for example, a personalized ad placement application, where the content owner observes the individual visitors to its website; and hence, has the context information, but must convey the ads that must be shown to each visitor to a separate entity that manages the marketing content. This will require communicating hundreds or thousands of adds to be placed at each round, from among a large set of possible adds, within the communication resource and delay constraints of the underlying communication channel, which is quantified as the number of bits available per agent. This problem may arise in other similar applications of CMABs with communication constraints between the decision-maker and the controller (Bouneffouf & Rish, 2019).

## 1.1 RELATED WORK

Given the amount of data that is generated by machines, sensors and mobile devices, the design of distributed learning algorithms is a hot topic in the ML literature. These algorithms often impose communication constraints among agents, requiring the design of methods that would allow efficient representation of messages to be exchanged. While rate-distortion theory deals with efficient lossy transmission of signals (Cover & Thomas, 2006b), in ML applications, we typically do not need to reconstruct the underlying signal, but make some inference based on that. These applications can be modeled through distributed hypothesis testing (Berger, Sep. 1979; Ahlswede & Csiszár, 1986) and estimation (Zhang et al., 2013; Xu & Raginsky, 2017) problems under rate constraints.

In parallel to the theoretical rate-distortion analysis, significant research efforts have been invested in the design of practical data compression algorithms, focusing on specific information sources, such as JPEG and BPG for image compression, or MPEG and H.264 for video compression. While adapting these tools to specific inference tasks is difficult, recently deep learning techniques have been employed to learn task-specific compression algorithms (Torfason et al., 2018; Jankowski et al., 2021), which achieve significant efficiency by bypassing image reconstruction.

While the above mainly focus on the inference task through supervised learning, here we consider the CMAB problem. There is a growing literature on multi-agent Reinforcement Learning (RL) problems with communication links (Foerster et al., 2016; Sukhbaatar et al., 2016; Havrylov & Titov, 2017; Lazaridou et al., 2017). These papers consider a multi-agent partially observable Markov decision process (POMDP), where the agents collaborate to resolve a specific task. In addition to the usual reward signals, agents can also benefit from the available communication links to better cooperate and coordinate their actions. It is shown that communication can help overcome the inherent non-stationarity of the multi-agent environment. Our problem can be considered as a special case of this general RL formulation, where the state at each time is independent of the past states and actions. Moreover, we focus on a particular setting in which the communication is one-way, from the decision-maker that observes the state and the reward, towards the controller that takes the actions. This formulation is different from the existing results in the literature involving multi-agent Multi-Armed Bandit (MAB). In Agarwal et al. (2021), each agent can pull an arm and communicate with others. They do not consider the contextual case, and focus on a particular communication scheme, where each agent shares the index of the best arm according to its experience. Another related formulation is proposed in Hanna et al. (2021), where a pool of agents collaborate to solve a common MAB problem with a rate-constrained communication channel from the agents to the server. In this case, agents observe their rewards and upload them to the server, which in turn updates the policy used to instruct them. In Park & Faradonbeh (2021), the authors consider a partially observable CMAB scenario, where the agent has only partial information about the context. However, this paper does not consider any communication constraint, and the partial/ noisy view of the context is generated by nature. Differently from the existing literature, our goal here is to identify the fundamental information theoretic limits of learning with communication constraints in this particular scenario.

## 2 PROBLEM FORMULATION

### 2.1 CONTEXTUAL MULTI-ARMED BANDIT (CMAB) PROBLEM

We consider $N$ agents, which experience independent realizations of the same CMAB problem. The CMAB is a sequential decision game in which the environment imposes a probability distribution $P_S$ over a set of contexts, or states, $\mathcal{S}$, which is finite in our case. The game proceeds in rounds, and at each round $t = 1, \ldots, T$, a realization of the state $s_{t,i} \in \mathcal{S}$ is sampled from the distribution $P_S$ for each agent $i \in \{1, \ldots, N\}$. At each time step $t$, and for each agent $i$, states are sampled iid according to $P_S$. In the usual CMAB setting, the decision-maker would observe the states $\{s_{t,i}\}_{i=1}^N$ of the agents, and choose an action (or arm) $a_{t,i} \in \{1, \ldots, K\} = \mathcal{A}$, for each agent, where $K$ is the total number of available actions, with probability $\pi_{t,i}(a_{t,i}|s_{t,i})$ given by a (possibly) stochastic policy $\pi_t : \mathcal{S} \to \Delta_K$. Here $\Delta_K$ is the K-dimensional simplex, containing all possible distributions over the set of actions. Once the actions have been taken by all the agents, at each round $t$ the environment returns rewards for all the agents following independent realizations of the same reward process, $r_{t,i} = r(s_{t,i}, a_{t,i}) \sim P_R(r|s_{t,i}, a_{t,i}), \quad \forall i \in \{1, \ldots, N\}$, which depends on the state and the action of the corresponding agent. Based on the history of returns up to round $t - 1$, the decision-maker can optimize its policy to maximize the sum of the rewards $G = \sum_{t=1}^T \sum_{i=1}^N r_{t,i}$.

### 2.2 RATE-CONSTRAINED CMAB

In our modified version, the process of observing the system states is spatially separated from the process of taking the actions. The environment states, $\{s_{t,i}\}_{i=1}^N$, are observed by a central entity, called the decision-maker, that has to communicate to the controller over a rate-constrained communication channel, at each round $t$, the information about the actions $\{a_{t,i}\}_{i=1}^N$ the agents should take. The decision-maker can exploit the knowledge accumulated from the states and rewards observed by all the agents up to round $t - 1$, denoted by $H(t - 1) = \left\{ (\{s_{1,i}\}_{i=1}^N, \{a_{1,i}\}_{i=1}^N, \{r_{1,i}\}_{i=1}^N), \ldots, (\{s_{t-1,i}\}_{i=1}^N, \{a_{t-1,i}\}_{i=1}^N, \{r_{t-1,i}\}_{i=1}^N), \right\} \in \mathcal{H}^{(t-1)}$, to optimize the policy to be used at round $t$. Consequently, the problem is to communicate the action distribution, i.e., the policy $\pi_t(a|s_t)$, which depends on the specific state realizations observed in round $t$, to the controller within the available communication resources while inciting the minimal impact on the performance of the learning algorithm.

Specifically, the decision-maker employs function $f_t^{(N)} : \mathcal{H}^{(t-1)} \times \mathcal{S}^N \to \{1, 2, \ldots, B\}$ to map the history up to time $t$ and the states of the agents at time $t$ to a message index to be transmitted over the channel. The controller, on the other hand, employs a function $g_t^{(N)} : \{1, 2, \ldots, B\} \to \mathcal{A}^N$ to map the received message to a set of actions for the agents. In general, both functions $f_t^{(N)}$ and $g_t^{(N)}$ can be stochastic. Average per period regret achieved by sequences $\left\{ f_t^{(N)}, g_t^{(N)} \right\}_{t=1}^T$ is given by

$$\rho^{(N)}(T) = \frac{1}{T} \mathbb{E} \left[ \sum_{t=1}^T \sum_{i=1}^N r(s_{t,i}, a^*(s_{t,i})) - r(s_{t,i}, a_t) \right], \tag{1}$$

where $a_{t,i} = g_{t,i}(m(t))$ is the action taken by agent $i$ based on message $m(t) = f_t^{(N)} \left( H(t - 1), \{s_{t,i}\}_{i=1}^N \right)$ transmitted in round $t$, $a^*(s_{t,i})$ is the action with maximum mean reward in state $s_{t,i}$, i.e., the optimal action, and the expectation is taken with respect to the state distribution $P_S$ and reward distribution $P_R$. We say that, for a given time period $T$ and $N$ agents, an average regret- communication rate pair $(\rho, R)$ is *achievable* if there exist functions $\left\{ f_t^{(N)}, g_t^{(N)} \right\}_{t=1}^T$ as defined above with rate $\frac{1}{N} \log_2 B \leq R$ and regret $\rho^{(N)}(T) \leq \rho$.

If a sufficiently large rate is available for communication, i.e., $R \geq \log K$, then the intended action for each agent can be reliably conveyed to the controller. Otherwise, to achieve the learning goal while satisfying the rate constraint, the decision-maker must apply a lossy compression scheme, such that the action distribution adopted by the pool of agents resembles the intended policy as much as possible.

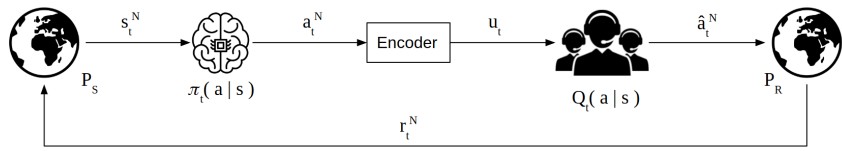

Figure 1: The problem of communicating policies.

## 3 SOLUTION

In this section, we present the proposed solution to the problem. First of all, we briefly discuss the algorithm adopted by the decision-maker to solve the CMAB. Then, the communication scheme for the RC-CMAB with a particular distortion function is provided, which is inherited by the learning task. We then identify two distinct rate regions resulting in linear and sub-linear regret, and propose a more practical coding scheme.

### 3.1 THOMPSON SAMPLING

In the proposed solution to the CMAB problem, the decision-maker adopts Thompson Sampling (TS) Thompson (1935). In particular, it makes use of one TS instance for each state $s \in \mathcal{S}$. Consequently, the decision-maker maintains an estimate of the distribution $p_t^{s,a}(\mu)$ of the mean reward $\mu^{s,a} \in \mathcal{D} \subseteq \mathbb{R}$ of action $a$ in state $s$ at time $t$. To take the decision in state $s_t$, the decision-maker would sample $\mu_t^{s_t,a} \sim p_t^{s_t,a}$, $\forall a \in \mathcal{A}$, and takes the action $a^* = \arg\max_{a \in \mathcal{A}} \{\mu_t^{s_t,a}\}$. This procedure is repeated for each agent $i \in \{1, \ldots, N\}$. After receiving the rewards $\{r_{t,i}\}_{i=1}^N$, the decision-maker can update its belief on $\mu^{s,a}$, i.e., the probabilities $p_t^{s,a}(\mu)$, in order to minimize the regret. We notice that this strategy induces a probability distribution $\pi_t(a|s)$ over the actions that is $\pi_t(a|s) = \int_{\mathcal{D}} p_t^{s,a}(\mu) \prod_{j=1, j \neq a}^K P_t^{s,j}(\mu) d\mu$, where $P_t^{s,j}(\mu)$ is the Cumulative Distribution Function (CDF) of $\mu_j$, where the random variables $\mu^{s,a}$ are considered independently distributed. However, the constraint on the rate imposed by the RC-CMAB formulation makes it infeasible for the decision-maker to sample, through the pool of agents, the actions directly from the true distribution $\pi_t(a|s)$. The agents have to use a proxy $Q_t(a|s)$, which is the one obtained from the message received. This problem is similar to approximate TS, where a proxy distribution is used to sample the actions, or the reward means, given that the true distribution is too complex to sample from. In that case, the bottleneck is given by the complexity of the mean reward distributions, whereas in this work, it is imposed by the limited-rate communication channel between the decision-maker and the controller.

### 3.2 OPTIMAL SOLUTION FOR THE RC-CMAB

We model the environment as a Discrete Memoryless Source (DMS), that generates states from a finite alphabet $\mathcal{S}$ with probability $P_S$, emitting sequences of $N$ symbols $s^N = (s_1, \ldots, s_N)$, one per agent. We then denote with $\hat{Q}_{s^N}(m)$ the empirical probability of state $m$ in $s^N$. We also consider the sequence of actions $a^N$, and denote with $\hat{Q}_{z^N}(m, j)$ the empirical joint probability of the pair $(m, j)$ in $z^N = ((s_1, a_1), \ldots, (s_N, a_N))$. The whole picture can be seen in Fig. 1. In the figure above, the actions taken by the agents are denoted by $\hat{a}$ to indicate that they can differ from $a$. However, we are interested in the probability distributions generating the sequences, thus we will denote with $A$ the random variables indicating both the actions at the controller and decision-maker side. We assume that the distribution $P_S$ is known (or accurately estimated).

The decision-maker can observe the realization $s^N$ of the contexts, and its task is to transmit an index $u \in \{1, \ldots, B\}$, and the agents can generate from $u$ a sequence $a^N$, such that $\hat{Q}_{s^N a^N}$ is close to $P_{SA}(sa) = P_S(s)\pi(a|s)$, where closeness depends on a distortion measure $\mathbb{E}[d(\hat{Q}_{S^N A^N}, P_{SA})]$, which in general is not an average of a per-letter distortion measure. The problem is a compression task in which the server has complete (or partial) knowledge of the states $s^N$, and wants to transmit a conditional probability distribution $\pi_{A|S}$ to the agents, consuming the minimum amount of bits, in such a way that the empirical distributions $\hat{Q}_{s^N a^N}$ given by the sequence induced by the agents

is close to the joint distribution $P_{SA}$ induced by the policy. For a distortion function $d(Q_{SA}, P_{SA})$ that is 1) nonnegative, 2) upper bounded by a constant $D_{max}$, 3) continuous in $Q_{SA}$, and 4) convex in $Q_{SA}$, in Kramer & Savari (2007), the authors provide the rate-distortion function $R(D)$, i.e., the minimum rate $R = \frac{\log_2 B}{N}$ bits per symbol such that $\mathbb{E}[d(\hat{Q}_{S^N A^N}, P_{SA})] \leq D$, in the limit when $N$ is arbitrarily large. The solution is given by

$$R(D) = \min_{Q_{A|S}:d(Q_{SA}, P_{SA}) \leq D} I(S; A), \tag{2}$$

where $Q_{SA} = P_S Q_{A|S}$ is the joint probability induced by the environment distribution $P_S$ and policy $Q_{A|S}$, which depends on the information sent by the decision-maker. As we can see, in the asymptotic limit of $N$ agents, the problem admits a single-letter solution, which also serves as a lower bound on the finite agent scenario. The studied RC-CMAB model described in Sec. 2.2 fits into this framework, and in the following section we identify an appropriate distortion function and provide its characterization.

### 3.3 THE KL-DIVERGENCE AS DISTORTION FUNCTION

When applying TS to the RC-CMAB problem with limited communication rate, the decision-maker may not be able to induce the controller to take samples from the true policy $\pi_t^{s,a}$. In Phan et al. (2019), the authors provide some theoretical guidelines to construct approximate sampling policies to make the posteriors, i.e., $p_t^{s,a}$, concentrate achieving sub-linear regret. In particular, they studied the case in which the sampling distribution $Q$ differs from the target posterior $\pi$, using $D_\alpha(\pi, Q)$ as distortion measure, which denotes the $\alpha$-divergence between the two, and is defined as

$$D_\alpha(\pi, Q) = \frac{1 - \int \pi(x)^\alpha Q(x)^{1-\alpha} dx}{\alpha(1 - \alpha}. \tag{3}$$

In Phan et al. (2019), it is shown in Theorem 1 that, for $\alpha > 0$, the condition $D_\alpha(\pi, Q) < \delta$ with $\delta > 0$, cannot guarantee that the posterior $\pi$ converges in sub-linear time, even for very small values of $\delta$. On the contrary, for $\alpha \leq 0$, the authors provide a scheme that can guarantee sub-linear regret. In particular, they suggest to introduce an exploration term $\rho_t \in o(1)$ such that $\sum_{t=1}^\infty \rho_t = \infty$, and the actions are sampled from $Q_t$ with probability $1 - \rho_t$, and uniformly at random with probability $\rho_t$, while $D_\alpha(\pi_t, Q_t) < \delta$ for all $t$. In this case, it is possible to obtain a sub-linear regret. Consequently, in order to solve the proposed RC-CMAB problem, we decide to investigate this strategy with $\alpha = 0$, which leads to the reverse KL-divergence from $Q$ to $\pi$, or equivalently, the KL-divergence from $\pi$ to $Q$, defined as $D_{KL}(Q, \pi) = \sum_{x \in \mathcal{X}} Q(x) \log \frac{Q(x)}{\pi(x)}$, when $x$ takes values in the discrete set $\mathcal{X}$. Consequently, to find the optimal constrained policy $Q_t(a|s)$, we need to find a solution to Eq. (2), which is a rate-distortion optimization problem, when the distortion function is given by the reverse KL-divergence.

We can rewrite the optimization objective of Eq. (2) as a double minimization problem (Sec. 10.8, (Cover & Thomas, 2006a))

$$R(D) = \min_{\tilde{Q}(a)} \min_{Q_{A|S}:d(Q_{SA}, P_{SA}) \leq D} \sum_{s,a} P(s)Q(a|s) \log_2 \frac{Q(a|s)}{\tilde{Q}(a)}. \tag{4}$$

Following (Lemma 10.8.1, (Cover & Thomas, 2006a)), the marginal $Q^*(y) = \sum_x P(x)Q(y|x)$ has the property

$$Q^*(y) = \arg \min_{\tilde{Q}(y)} D_{KL}(P(x)Q(y|x)||P(x)\tilde{Q}(y)), \tag{5}$$

that is, it minimizes the KL-divergence between the joint and the product $P(x)Q(y)$, $\quad \forall Q \in \Delta_K$. This means that $\tilde{Q}(a)$ obtained by solving Eq. (4) is indeed the marginal over the actions induced by $Q(a|s)$. Exploiting this formulation, it is possible to apply the iterative *Blahut-Arimoto algorithm* to solve the problem and find the solution. In particular, given that the two sets $\Delta_K$ and $\Delta_S$ are made of probability distributions, and the target measure is the KL-divergence, the algorithm does converge to the minimum (Csiszár & Tusnády, 1984). The process is initialized by setting a random $\tilde{Q}_0(a)$, which is used as a fixed point to compute

$$Q_1^*(a|s) = \text{argmin}_{Q_{A|S}:d(Q_{SA}, P_{SA}) \leq D} \sum_s P(s) \sum_a Q(a|s) \log_2 \frac{Q(a|s)}{\tilde{Q}_0(a)}. \tag{6}$$

From $Q_1^*(a|s)$, we compute the optimal $Q_1^*(a)$ by solving Eq. (5), which is simply the marginal $Q_1^*(a) = \sum_s P(s)Q_1^*(a|s)$. The process is iterated until convergence.

We now solve the inner minimization problem, i.e., Eq. (6) with fixed $\tilde{Q}(a)$. As we can see, the constraint on the distortion tries to keep $Q_{A|S}$ close to $\pi_{A|S}$, which is the target distribution. By minimizing the mutual information, we want $Q_{A|S}$ as close to $Q_A$ as possible, which is the marginal known by the agent (it is sent once at the beginning of the round and does not depend on the state realizations), in order to reduce the number of necessary bits to be transmitted. The solution to Eq. (2) is a conditional distribution $Q_{A|S}$ that is a combination of the target policy conditioned on $S$, and the marginal distribution.

We now discuss some simple corner cases. If the maximum acceptable distortion is 0, we need $Q_{A|S} = \pi_{A|S}$, thus the rate is equal to $\mathbb{E}_{P_S}\left[D_{KL}\left(\pi_{A|S}||\pi_A\right)\right]$. If $S$ and $A$ are independent, i.e., the policy does not depend on the state, we have $\pi_{A|S} = \pi_A$, thus the decision-maker does not need to send any bits to the agents, which can just sample $A \sim \pi_A$. Moreover, if we can find a distribution $Q_A$ such that $\mathbb{E}_{P_S}\left[D_{KL}\left(Q_A||\pi_{A|S}\right)\right] \leq D$, then the best strategy is not to convey any message to the controller, as the constraint is already satisfied with rate $R(D) = 0$. The problem is to find, among all conditional distributions $Q(a|s)$ that satisfy the constraint, the one with the minimum divergence from $Q(a)$, in order to minimize the rate $\frac{\log_2 B}{N}$, i.e., the number of bits to be consumed per agent, for an arbitrarily large $N$, in the general case. We solve the problem using the Lagrangian multipliers, and obtain the shape of the optimal distribution given by

$$Q_{\gamma^*}(a|s) = \frac{\tilde{Q}(a)^{\gamma^*} P(a|s)^{1-\gamma^*}}{\sum_{a' \in \mathcal{A}} \tilde{Q}(a')^{\gamma^*} P(a'|s)^{1-\gamma^*}}, \quad \forall s \in \mathcal{S}, a \in \mathcal{A}, \tag{7}$$

where $\gamma^*$ is s.t. $\mathbb{E}_{P_S}[D_{KL}(Q_{\gamma^*}||P)] = D$. The derivation is provided in Appendix A.1.

### 3.4 Asymptotic Regret

To prove the results on the achievable regret, we need additional assumptions, which are contained in the Assumption 1 in Russo (2016), which states that rewards have to be distributed following canonical exponential families, and the priors used by TS over the average rewards have to be uniformly bounded, i.e., bounded away from zero $\forall(s,a)$. The proofs of both Lemmas are reported in the Appendix D.

In the following, we provide the minimum rate needed to achieve sub-linear regret in all states $s \in \mathcal{S}$. We now define $H(A^*)$ as the marginal entropy of the optimal arm, computed based on the marginal $\pi^*(a) = \sum_s P_S(s)\pi^*(a|s)$, and defined as $H(A^*) = \sum_a \pi^*(a) \log \frac{1}{\pi^*(a)}$, and we prove that it is the minimum rate required to achieve sub-linear regret.

**Lemma 3.1.** *If $R < H(A^*)$, then it is not possible to convey a policy $Q_t(s,a)$ that achieves sub-linear regret in all states $s \in \mathcal{S}$.*

The following Lemma provides the achievibility part.

**Lemma 3.2.** *If $R > H(A^*)$, then achieving sub-linear regret is possible in all states $s \in \mathcal{S}$.*

The consequence of this second Lemma is that, even if the exact TS policy $\pi_t$ cannot be transmitted $\forall t$, as long as sufficient rate is available, i.e., $R > H(A^*)$, it is still possible to achieve sub-linear regret. Following the notation introduced in Sec. 2.2, this implies that, as $T \to \infty$, and for all sub-linear regrets $\rho$, the regret-communication pair $(\rho, R)$ is achievable as long as $R > H(A^*)$. Moreover, we argue that the policy construction found in Sec. 3.3 can achieve better empirical performance w.r.t. the scheme used to prove Lemma 3.2.

### 3.5 Practical Coding Scheme

The above analysis allows us to characterize an information theoretical bound on the optimal performance, but does not provide a constructive communication scheme. To find a practical coding scheme, we propose a solution that is based on state reduction and computes a compact state representation. In essence, the decision-maker constructs a message containing the new state representations $\hat{s}(s) \in \hat{\mathcal{S}}$ of $s$, one for each agent, and send it over the channel. Once the agents have received

the message, they can sample the actions according to a common policy $Q_{\hat{s}}(a|\hat{s})$, which is defined on the compressed state space $\hat{\mathcal{S}}$. If the rate constraint imposes $B$ bits per agent, it means that it is possible to transmit at most $2^B$ different states to each agent. The idea is to group the states into $2^B = M$ clusters $\hat{s}_1, \ldots, \hat{s}_M$, minimizing $\sum_s P(s) D_{KL}(Q_{\hat{s}}(a|\hat{s})||\pi(a|s))$, where $Q_{\hat{s}}(a|\hat{s})$ is the policy defined on the state $\hat{s}(s) \in \hat{\mathcal{S}}$.

To find the clusters and relative policies we employ the well-known Lloyd algorithm, which is an iterative process to group states into $2^B$ clusters. First of all, knowing the policy $\pi$, the decision-maker maps each state $s_i$ to a K-dimensional point $\boldsymbol{\alpha^i} = \pi(\cdot|s_i) \in \Delta_K$, finding $|\mathcal{S}| = L$ different points $\boldsymbol{\alpha^1}, \ldots, \boldsymbol{\alpha^L}$. Then, it generates $2^B = M$ random points $\boldsymbol{\mu^1}, \ldots, \boldsymbol{\mu^M} \in \Delta_K$ as initial centroids, i.e., representative policies, and iterates over the following two steps:

1. Assign to each point $\boldsymbol{\alpha^i}$ the class $j^* \in \{1, \ldots, M\}$ such that $j^* = \arg\min_j D_{KL}(\boldsymbol{\mu^j}||\boldsymbol{\alpha^i})$, i.e., minimizing the divergence between the representative $\boldsymbol{\mu^{j^*}}$ and the original policy, which is the point $\boldsymbol{\alpha^i}$. For each cluster $j$, we now define with $\mathcal{S}^j$ the set containing the states associated to the policies in the cluster.

2. Update $\boldsymbol{\mu^1}, \ldots, \boldsymbol{\mu^M}$ such that $\boldsymbol{\mu^j} = \arg\min_{\boldsymbol{\mu} \in \Delta_K} \sum_{s \in S_j} P(s) D_{KL}(\boldsymbol{\mu}||\pi(\cdot|s))$, which is still a convex optimization problem, and can be solved again applying the Lagrangian multipliers. The solution is

$$\boldsymbol{\mu^j} = \frac{\prod_{s \in \mathcal{S}_j} \pi(\cdot|s)^{\frac{P(s)}{A(\mathcal{S}_j)}}}{Z}, \tag{8}$$

where the product has to be considered element-wise, $A(\mathcal{S}_j)$ is the sum of the probabilities of states in $\mathcal{S}_j$, i.e., $A(\mathcal{S}_j) = \sum_{s \in \mathcal{S}_j} P(s)$, and $Z$ is the normalizing factor. After computing the new centroids, we go back to step (1). The derivation of Eq. (8) is provided in Appendix A.2.

The process continues until the new solution does not decrease the KL-divergence between the clustered and the target policy $D_{KL}(Q_{\hat{s}}(\hat{s}, a)||P(s, a)) = \sum_{j=1}^M \sum_{s \in \mathcal{S}_j} P(s) D_{KL}(\boldsymbol{\mu^j}||\pi(\cdot|s))$.

**Observation** Note that the controller is assumed to know the $2^{NR}$ policies from which it samples the actions of the agents. This can be transmitted at the beginning of each round. In this case, the scheme is efficient as long as $N \log_2 K >> BL\lambda \log_2 K$, where $\lambda$ is the number of bits used to represent the values of the Probability Mass Function (PMF) $Q_{\hat{s}}(\cdot|\hat{s})$. For this reason, we provide a scheme where the new policy is updated not at every transmission, but just when the new target $\pi$ has changed considerably. In particular, if we denote with $\pi^{cls}$ the policy defined over the compressed state representation, with $\pi^{last}$ the last policy used to compute $\pi^{cls}$, and with $\pi$ the updated target policy, we compute and transmit $\pi^{cls}$ every time $D_{KL}(\pi^{last}||\pi) > \beta$.

## 4 NUMERICAL RESULTS

In this section, we provide numerical results in support of our theoretical analysis. Additional experiments and details can be found in Appendix C

### 4.1 RATE-DISTORTION FUNCTION

In this first experiment, we analyze the rate-distortion function, and the related optimal $Q(a|s)$, when the distortion is given by the KL-divergence in three different problems, when the numbers of states, $L$, and of actions, $K$, are both equal to 16. The target policies $\pi(a|s)$ have a one-to-one relation with the states, and are generated such that $\pi(a = i|s = j) = 0.99$ if $j = i$, and uniformly distributed otherwise. We refer to this as the *Deterministic* case. In the second experiment, the target policies are generated in a similar way, but the other actions' probabilities take random values in $(0, 0.05]$, and then normalized. This, we call the *Random Deterministic* case. In the third experiment, the states are grouped in 4 blocks, such that $\pi(a = i|s = j) = 0.99$ if $\lfloor j/4 \rfloor = i$, and uniformly distributed otherwise. This case is denoted as the *Block* one. In all three experiments, the distribution $P_S$ is uniform over the state space $\mathcal{S}$. The rate-distortion curves are reported in Fig. 2a. As we can see, in the *Deterministic* case, the point with zero distortion has $R(0) \sim H(P_S)$, where $H(P_S)$ is

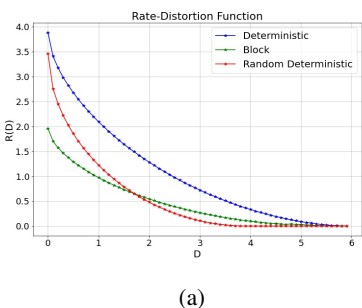

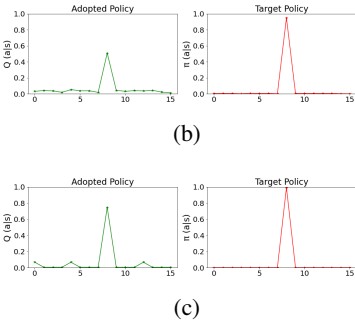

Figure 2: Rate-Distortion function for the 3 different experiments (a). $\pi(a|s = 8)$ and $Q(a|s = 8)$ in the third experiment, when the rate $R$ is equal to 1 (b). $\pi(a|s = 8)$ and $Q(a|s = 8)$ in the second experiment, when the rate $R$ is equal to 1.

the Shannon entropy of $P_S$, as expected. Indeed, in this case, the action distributions are strongly correlated with the state, thus we need an accurate knowledge of it. In the second experiment, the starting point is similar, but the curve decreases more rapidly, caused by the random values of the other actions' probabilities. In the third experiment, the zero distortion point is achieved at $R(0) \sim 2$ bits, since, similarly to the *Deterministic* experiment, action probabilities are correlated with the state realization, but are grouped into four different cases. Again, given the uniform distribution over the state, $R(0) \sim \text{H}(\text{Unif}_{[4]})$, where $\text{Unif}_{[4]}$ is the uniform distribution over a set of 4 elements. In Fig. 2c, the approximate distributions for state $s = 8$ are reported for the *Deterministic* and *Random Deterministic* cases, when $R = 1$.

## 4.2 CONTEXTUAL MULTI-ARMED BANDIT

We now analyze the RC-CMAB problem presented in Sec. 2, and apply the clustered policy schemes to solve it. In particular, we compare the performance of the *Perfect* agent, which applies TS without any rate constraint, thus admits samples from the true posterior $\pi$, with the performance of the *Comm*, *Cluster*, and *Marginal* agents. The *Comm* agent uses the optimal scheme provided in Sec. 3.2, the *Cluster* agent implements the practical coding scheme provided in Sec. 3.5, with $B = \lceil R \rceil$ bits per agent, where $R$ is the rate adopted by the *Comm* agent; the *Marginal* agent adopts the marginal over the states, computed from the target policy $\pi$ and the environment distribution $P_S$, and serves as a lower bound on the performance. State distribution $P_S$ is uniform over $L = 16$ states, and there are $K = 16$ actions, $N = 100$ agents, and the total number of rounds is $T = 200$. The threshold for changing the clustered policy is set to $\beta = 0.2$, and the communication rate constraint to $R = 2.5$. In this experiment, for each state $s_i \in \mathcal{S}$, the best reward is given by the arm $a_j$ such that $i = j$, with $i, j \in \{0, \dots, 15\}$. In particular, the reward behind arm $i$ when in state $j$ is a Bernoulli random variable with parameter $\mu_j = 0.8$ if $i = j$, whereas $\mu_j \sim \text{Unif}_{[0,0.75]}$ if $i \neq j$. In this case, the best action response is strongly correlated with the state realization, thus a sufficiently high rate is required to sample from the target policy $\pi$. As it can be observed from Fig. 3(a), the theoretical rate to transmit the target policy is related to the amount of information the agent is learning from the environment, and it is computed using Eq. (2). Indeed, during the first $\sim 40$ iterations, the rate is below $R = 2.5$. As the learning process continues, the required rate for reliable transmission increases, as the mutual information between $S$ and $A$ increases. We highlight that, in order to fairly represent the 16 different actions, one would need 4 bits. Indeed, another way of looking at Eq. (2) is through *bottleneck principle* (Igl et al., 2019), which is used to constrain the mutual information between the states and the actions, and to encourage exploration, that can be exploited to reduce the required rate when training multi-agent systems.

In Fig 3 (b), the KL-divergence between the last policy used to compute the compressed state representation, i.e., $\pi^{last}$ in Sec. 3.5, and the target known by the decision-maker is reported. The trend shows that, at the beginning of the learning process, it oscillates rapidly, as the target policy is changing significantly between two iterations. In gray, the threshold $\beta = 2$ is highlighted, indicating when a new cluster policy has to be sent. In this experiment, the decision-maker had to send a new policy, on average, once every $\sim 7$ rounds. Fig. 4 (a) reports the regret at state $s = 10$ (the other

plots can be seen in Appendix C.2), for the four different agents, as defined in Eq. (1). We can see that both the *Cluster* and the *Comm* agents can achieve sub-linear regret in this particular state. In Fig. 4 (b), the average rewards obtained by the agents are reported, for the different states separately. We can see that the developed scheme is still able to achieve good performance.

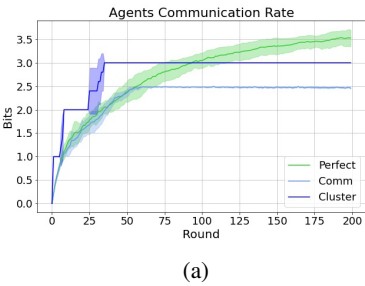

(a)

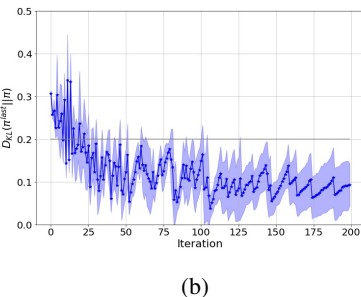

(b)

Figure 3: Asymptotic rates to convey the *Perfect* and *Comm* policies, and the bits used by the *Cluster* agent, averaged over 5 runs (a). $D_{KL}(\pi^{last}||\pi)$, averaged over 5 runs (b).

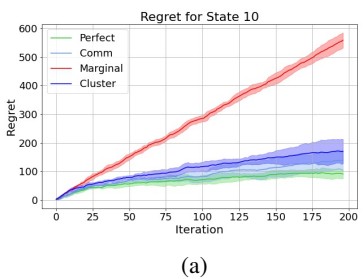

(a)

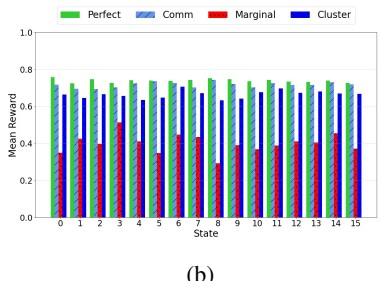

(b)

Figure 4: Cumulative regret for different agents, together with the performance of the $Marginal$ agent, evaluated for the state $s = 10$ (a). Average reward per state (b).

## 5 CONCLUSION

We have studied the RC-CMAB problem, in which an intelligent entity, i.e., the decision-maker, observes the contexts of $N$ parallel CMAB processes, and has to decide on the actions depending on the current contexts and the past actions and rewards. However, the actions are implemented by a controller that is connected to the decision-maker through a rate-constrained communication link. First, we cast the problem into the proper information-theoretic framework, formulating it as a policy compression problem, and provided the optimal compression scheme in the limit of an infinite number of agents, when the adopted distortion measure is the KL-divergence between the compressed policy adopted by the controller and the policy of the decision-maker. We then characterize the minimum needed rate to obtain sub-linear regret, and prove that any rates above this threshold can achieve it. In the end, we designed a practical coding scheme to transmit the actions for a finite $N$, which relies on a compressed state representation Finally, we evaluated the performances of the policy obtained through the asymptotic information theoretic formulation, and the one obtained through the clustering scheme, and observed a close gap between the two. We numerically showed the relation between the asymptotic rate bound and the learning phase of agents, showing how it is possible to save communication resources when training a multi-agent system like the one considered. We believe that this work can serve as a first step towards understanding the fundamental performance limits of multi-agent decision-making problems under communication constraints, and highlights the intimate relation between the communication scheme and the learning process. Ongoing work include deriving an information theoretic converse result on the regret performance, and generalizing the framework to reinforcement learning problems.

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

## APPENDIX

## A   POLICIES DERIVATION

### A.1   OPTIMAL POLICY

To solve this problem, we solve the related Lagrangian

$$\mathcal{L}(Q(a|s), \lambda, \mu) = \sum_s P(s) \sum_a Q(a|s) \log \frac{Q(a|s)}{\tilde{Q}_a} + \lambda \left( \sum_s P(s) \sum_a Q(a|s) \log \frac{Q(a|s)}{P(a|s)} - D \right) +$$
$$+ \mu \left( \sum_s P(s) \sum_a Q(a|s) - 1 \right)$$

where the Lagrangian multiplier $\lambda$ has to be optimized to meet the constraints on the divergence, whereas $\mu$ ensures that the solution is a probability distributions, i.e., the elements sum up to one. The positivity constraints on the terms are already satisfied by the fact that the solution has an exponential shape. We first take the derivative of the Lagrangian w.r.t. to the terms $Q(a|s)$ and set it to zero

$$\frac{\partial \mathcal{L}(Q(a|s), \lambda, \mu)}{\partial Q(a|s)} = P(s) \log \frac{Q(a|s)}{\tilde{Q}(a)} + P(s) - \sum_{s'} P_{s'} Q(a|s') \frac{P(s)}{\tilde{Q}(a)} +$$
$$+ \lambda P(s) \left( \log \frac{Q(a|s)}{P(a|s)} + 1 \right) + \mu P(s) = 0$$

finding

$$\log \frac{Q(a|s)^{1+\lambda}}{\tilde{Q}(a) P(a|s)^\lambda} = -(\lambda + \mu)$$
$$Q(a|s)^{1+\lambda} = e^{-(\mu+\lambda)} \tilde{Q}(a) P(a|s)^\lambda$$
$$Q(a|s) = e^{\frac{-(\mu+\lambda)}{1+\lambda}} \tilde{Q}(a)^{\frac{1}{1+\lambda}} P(a|s)^{\frac{\lambda}{1+\lambda}}$$

and now we rewrite with $\gamma = \frac{1}{1+\lambda}$, $\gamma \in [0,1]$, and $\mu$ such that $e^{\frac{-(\mu+\lambda)}{1+\lambda}}$ is the normalizing factor. The solution has thus the shape

$$Q_\gamma(a|s) = \frac{\tilde{Q}(a)^\gamma P(a|s)^{1-\gamma}}{\sum_{a' \in \mathcal{A}} \tilde{Q}(a')^\gamma P(a'|s)^{1-\gamma}}, \quad \forall s \in \mathcal{S}, a \in \mathcal{A}. \tag{9}$$

By the convexity of KL-Divergence and its triangular inequality, we know the solution lies on the boundary of the constraints, i.e., when $\mathbb{E}_{P_S}[D_{KL}(Q_{\gamma^*}||P)] = D$. Consequently, if we can numerically find $\gamma^*$ s.t. $\mathbb{E}_{P_S}[D_{KL}(Q_{\gamma^*}||P)] = D$, we solve Eq. (6).

## A.2 UPDATE CENTROIDS

Again, we compute the optimal centroids by solving the Lagrangian

$$\mathcal{L}(\mu_a^j, \lambda) = \sum_{s \in \mathcal{S}_j} P(s) \sum_{a \in \mathcal{A}} \mu_a^j \log \frac{\mu_a^j}{\pi(a|s)} + \lambda \left( \sum_{a \in \mathcal{A}} \mu_a^j - 1 \right)$$

taking its derivative and solving the equality

$$\frac{\partial \mathcal{L}(\mu_a^j, \lambda)}{\partial \mu_a^j} = \sum_{s \in \mathcal{S}_j} P(s) \left( \log \frac{\mu_a^j}{\pi(a|s)} + 1 \right) + \lambda = 0$$

finding

$$\log \mu_a^j A(\mathcal{S}_j) = \sum_{s \in \mathcal{S}_j} P(s) \log \pi(a|s) + A(\mathcal{S}_j) + \lambda$$

$$\log \mu_a^j = \sum_{s \in \mathcal{S}_j} \frac{P(s)}{A(\mathcal{S}_j)} \log \pi(a|s) + 1 + \frac{\lambda}{A(\mathcal{S}_j)}$$

$$\mu^j = \frac{\prod_{s \in \mathcal{S}_j} \pi(\cdot|s)^{\frac{P(s)}{A(\mathcal{S}_j)}}}{Z},$$

where $Z$ is the normalizing factor, obtaining the shape expressed in Eq. (8).

## B    PRESENTED ALGORITHMS

In this appendix, the algorithms for the theoretically-optimal and cluster policies are described.

### B.1    THEORETICAL OPTIMAL POLICY

This is the pseudocode of the algorithm at the decision-maker side, when adopting the information-theoretic inspired compressed policy. The only task for the controller is to decode the received message, and to communicate the actions to the agents. We define with $[A]$ the set $\{1, \ldots, A\}$, and with $\text{Unif}_B$ the uniform probability over the set $B$.

---
**Algorithm 1** Optimal Policy
---
**Input** rate $R$, n° agents $N$, n° actions $K$ and state distribution $P_S$
**Initialize** Thompson Sampling policy $\pi$, and compressed policy $Q(\cdot|s)$ to $\text{Unif}_{[K]}$
**foreach** $t \in 1, \ldots, T$ **do**
    Observe $s_t^N = (s_{t,1}, \ldots, s_{t,N})$
    Sample $a_t^N = (a_{t,1}, \ldots, a_{t,N})$ from $Q(\cdot|s)$
    Code $a_t^N$ into $u_t = f_t(a_t^N)$ and transmit to the controller
    Observe $r_t^N = (r_{t,1}, \ldots, r_{t,N})$
    Update action posteriors $\pi(\cdot|s)$ using the **Thompson Sampling Algorithm**
    Update $Q(a|s)$ using the **Blahut-Arimoto Algorithm** (Iterate over Eq. 5 and Eq. 6)

---

### B.2    CLUSTER POLICY

This is the pseudocode of the algorithm at the decision-maker side, when adopting the cluster policy. The task for the controller is to sample, at time $t$ and for each agent $i$, the action $a_{t,i}$ according to the last received centroid $\boldsymbol{\mu}_{j(s_{t,i})}$, whose components convey the action probabilities.

---
**Algorithm 2** Cluster Policy
---
**Input** per-agent bit $B$, n° agents $N$, n° actions $K$ and state distribution $P_S$, threshold $\beta$
**Initialize** Thompson Sampling policy $\pi$
Compute the centroids $\boldsymbol{\mu_j}$ for $j = 1, \ldots, 2^B$ using $\pi$ with Lloyd algorithm and the rule in Eq. 8
Transmit the centroids $\boldsymbol{\mu_j}$ to all agents.
$\pi^{last} = \pi$
**foreach** $t \in 1, \ldots, T$ **do**
    Observe $s_t^N = (s_{t,1}, \ldots, s_{t,N})$
    **if** $D_{KL}\left(\pi^{last}||\pi\right) > \beta$ **then**
        Update the centroids $\boldsymbol{\mu_j}$ using $\pi$ with Lloyd algorithm and the rule in Eq. 8
        Transmit the centroids $\boldsymbol{\mu_j}$ to all agents.
    $\forall s_{t,i}$ compute the index $j(s_{t,i})$ indicating its belonging cluster
    Transmit the vector $j_t^N = (j(s_{t,1}), \ldots, j(s_{t,N}))$
    Observe $r_t^N = (r_{t,1}, \ldots, r_{t,N})$
    Update action posteriors $\pi(\cdot|s)$ using the **Thompson Sampling Algorithm**

---

## C EXPERIMENTS

In this section, additional experiments are provided to clarify the proposed scheme, and quantify the trade-offs between rate and regret.

### C.1 STATE REPRESENTATION

In Fig. 5 we report the policy clustering results when applying the coding scheme explained in Sec. 3.5, when $R = 2$ bits can be used to represent 20 randomly generated policies. In the figure, it is possible to see the 4 policies representative of the 4 compressed state representations. As we can see, the policy centroids found by the coding scheme try to fairly resemble the shapes of all the policies within the same cluster.

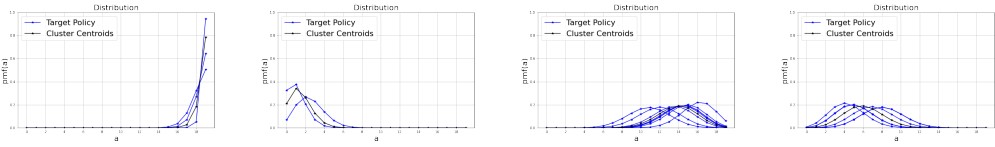

Figure 5: Clusters and their representatives with $L = 20$ and $b = 2$.

### C.2 DETERMINISTIC

In this section we reported some more details on the RC-CMAB experiments presented in Sec. 4.2.

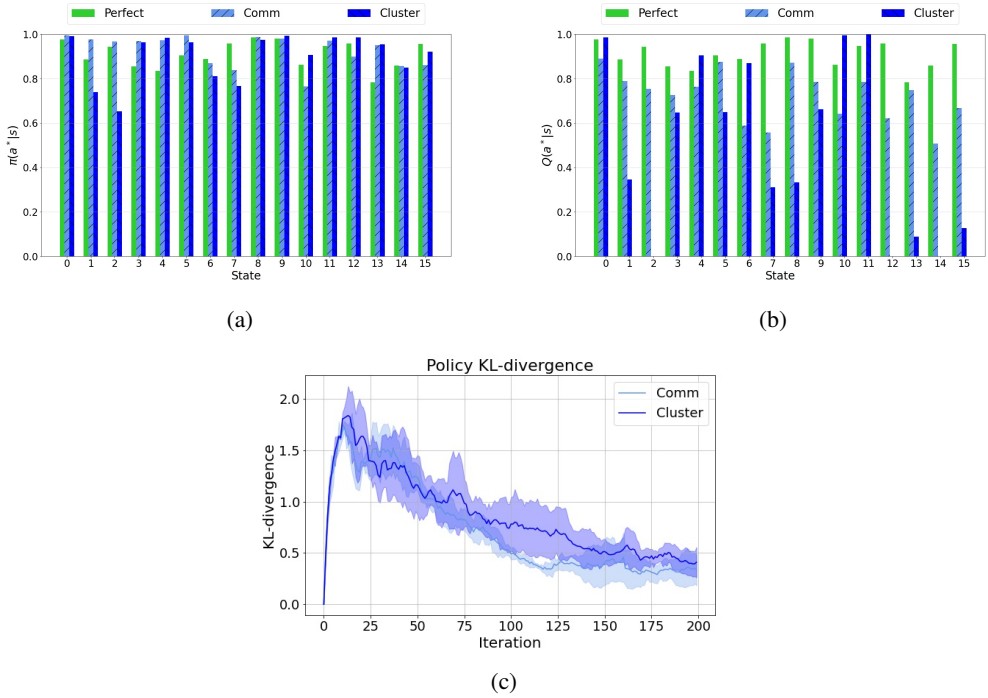

Figure 6: Best action probability for a given state, for the posterior $\pi(a^*|s)$ (a) and compressed policy $Q(a^*|s)$ (b) for the different agents. KL-divergence between the agents' action posteriors, and the target one (c).

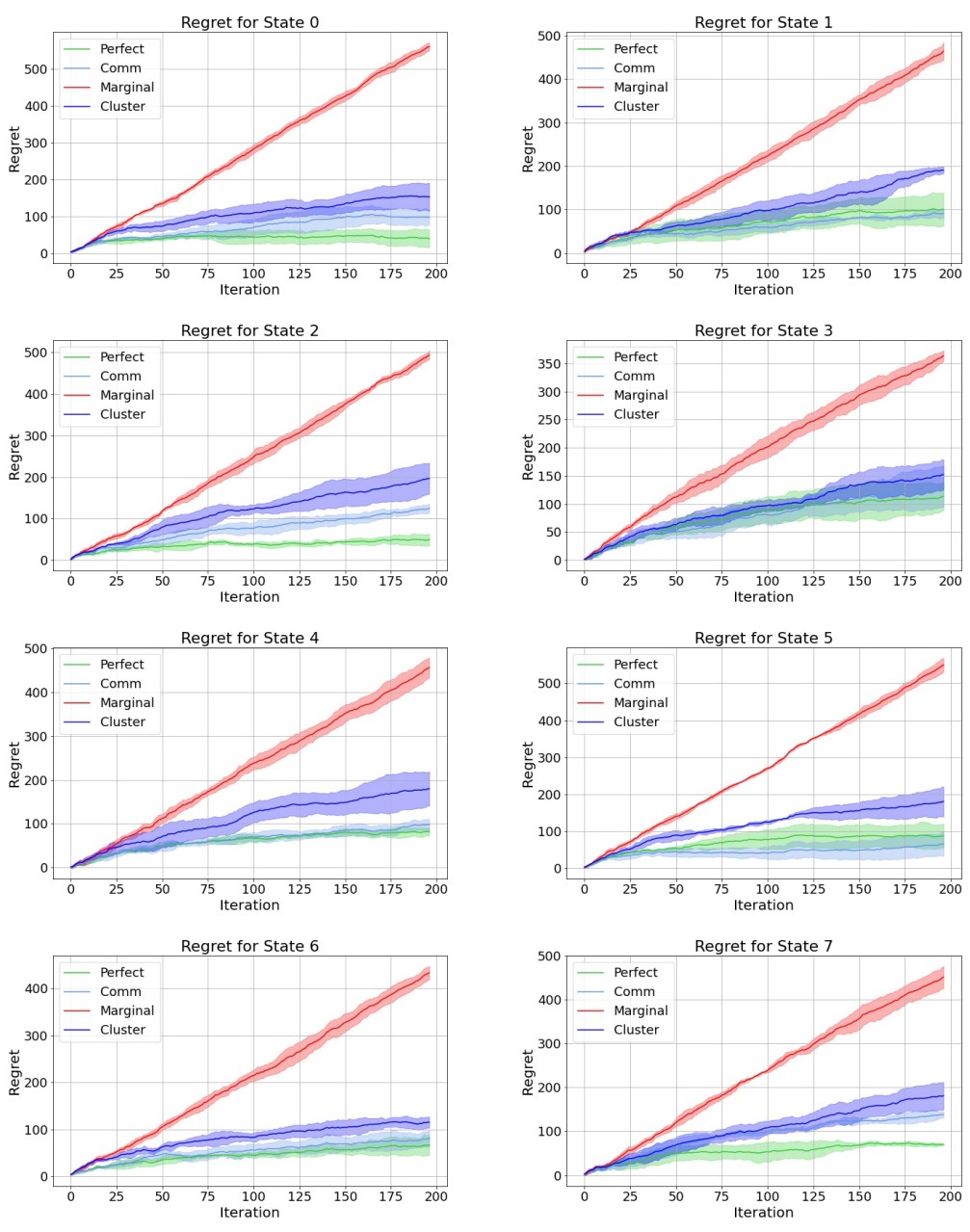

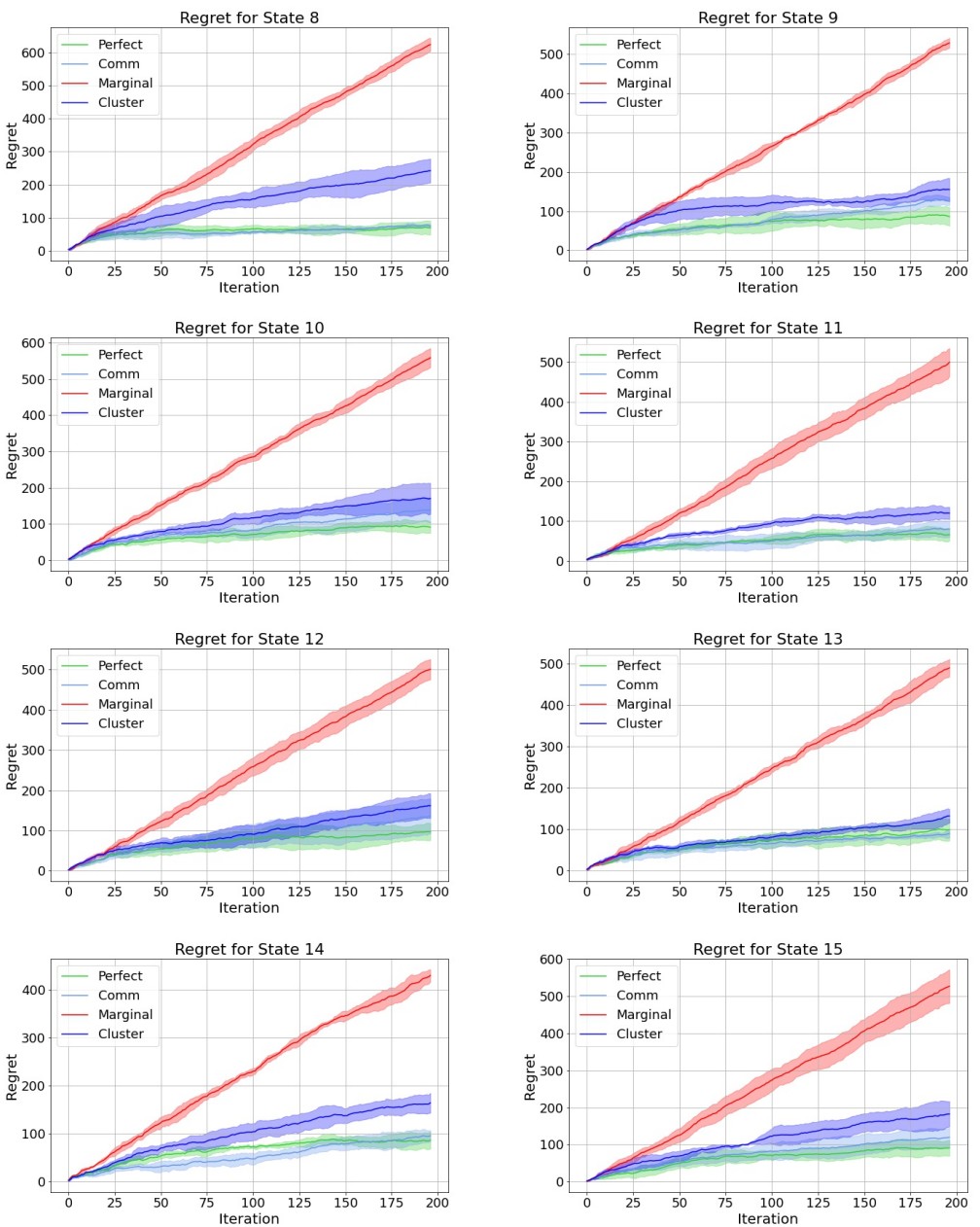

Figure 8: Cumulative average regret $\pm$ one std for the different agents, from state $s = 0$ to state $s = 15$

### C.3   8-BLOCK EXPERIMENT

In this second RC-CMAB experiment, the setting is similar to the one presented above, but the best action response is not a one-to-one mapping with the state. Again, $a \in \{0, \dots, 15\}$ and $s \in \{0, \dots, 15\}$, but the Bernoulli parameter $\mu_a(s)$ for action $a$ in state $s$ is 0.8 if $\lfloor \frac{s}{2} \rfloor = a$, and sampled uniformly in $(0, 0.75]$ otherwise. Thus, the best action responses are grouped into 8 different classes, depended on the state realization. In this case, the rate is limited to $R = 2$.

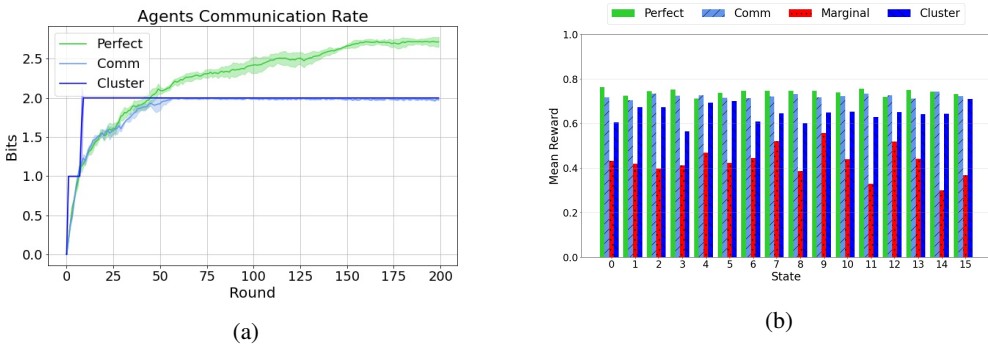

(a)

(b)

Figure 9: Block experiment : asymptotic rates to convey the *Perfect* and *Comm* policies, and the bits used by the *Cluster* agent, averaged over 5 runs (a). Average reward achieved in each state by the pool of agents (b)

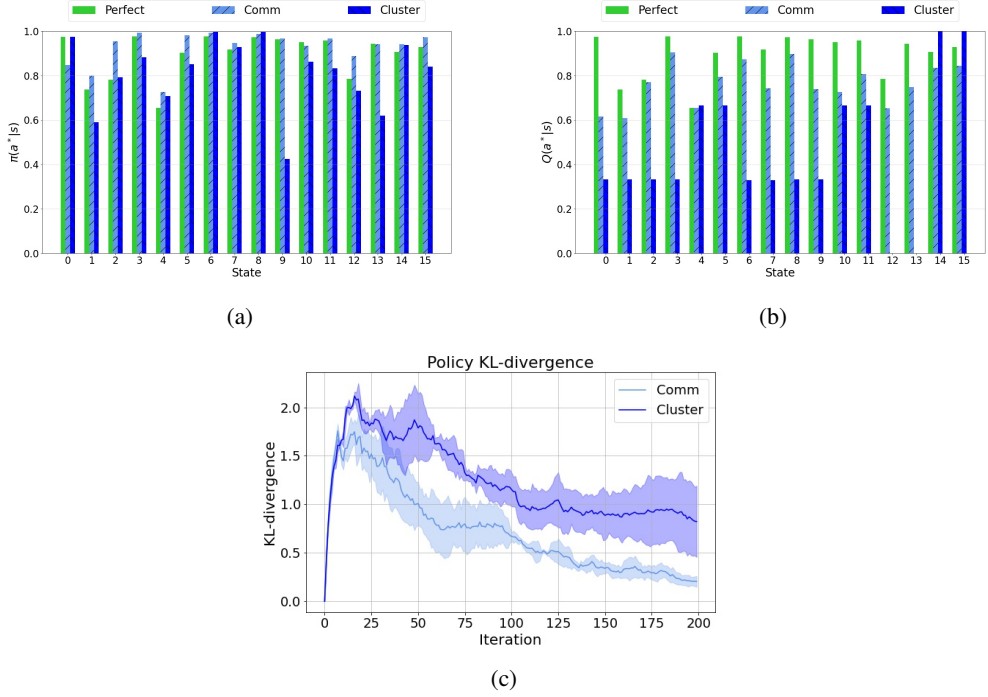

(a)

(b)

(c)

Figure 10: Block experiment : Best action probability for a given state, for the posterior $\pi(a^*|s)$ (a) and compressed policy $Q(a^*|s)$ (b) for the different agents. KL-divergence between the agents' action posteriors, and the target one (c).

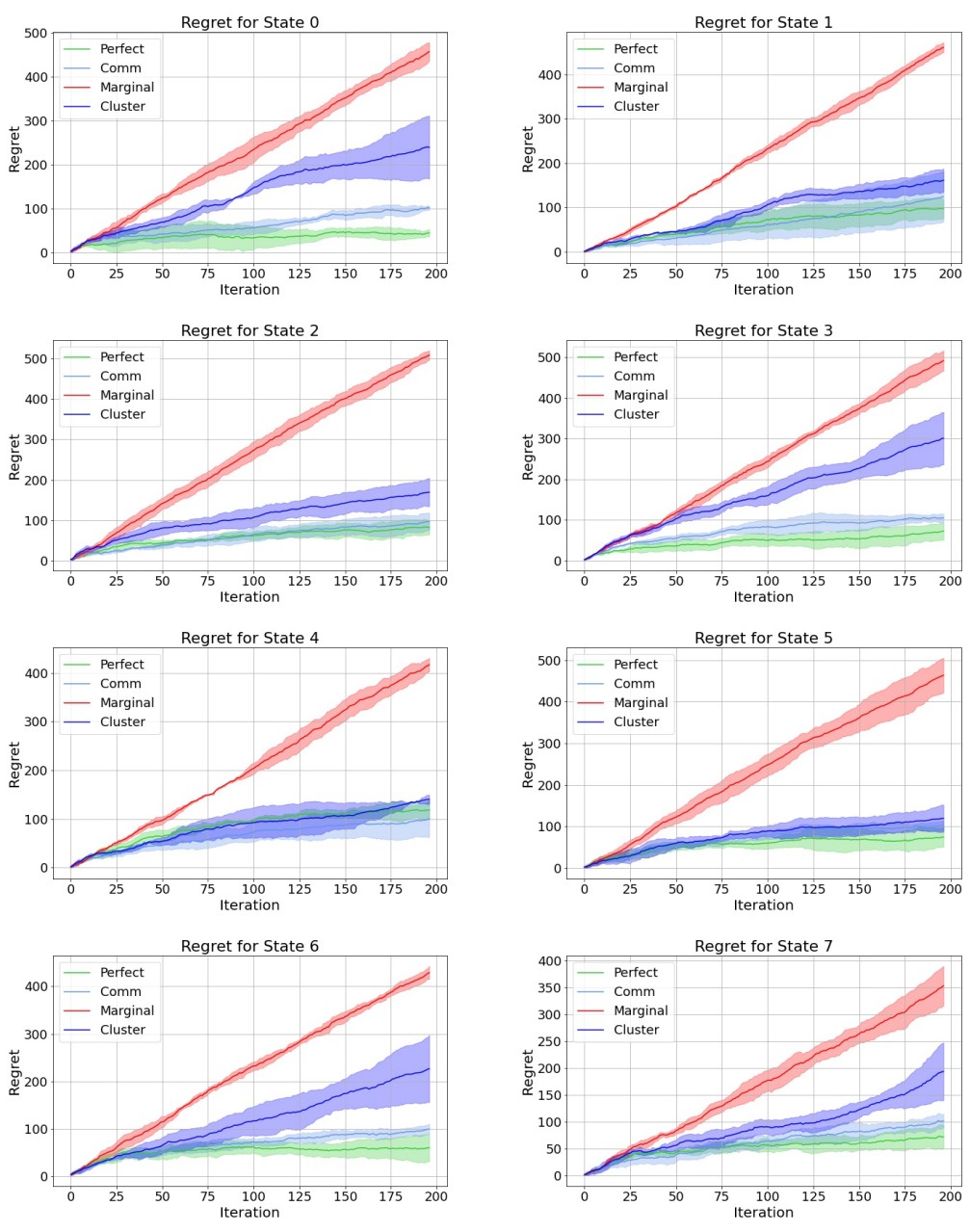

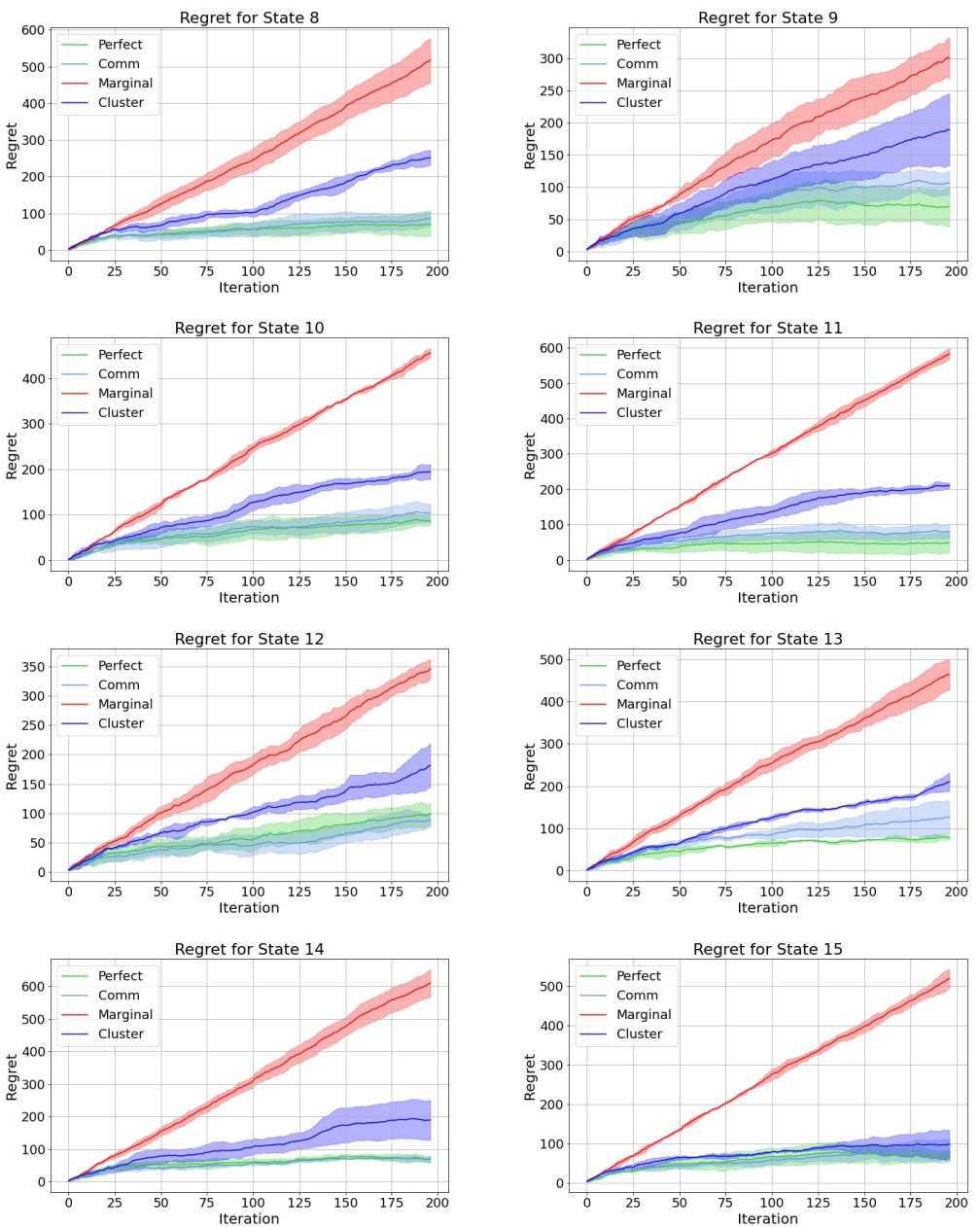

Figure 12: Block experiment : cumulative average regret $\pm$ one std for the different agents, from state $s = 0$ to state $s = 15$

## C.4 REGRET VS NUMBER OF CLASSES

In this last experiment, the environment is the same as the one described in Sec. 4.2, a part form the number of agents, that here is $N = 50$. The task is to quantify the effect of the number of per-agent bits $B$ in the cluster policy on the achievable regret, which varies from $1$ to $4$. As we can see from the plots below, when $B$ is equal to $1$ or $2$, which in turn means that the number of clusters is, respectively, $2$ and $4$, the regret is not sub-linear in several states. However, the policy with just $2$ bits can achieve performance comparable with the $3$ and $4$ bits policies in some states, e.g., state $4$ and $6$. This is due to the fact that even in those cases with not enough bits, the posteriors still converges to good solutions, thus if a cluster contains, for example, one state, it will convey the best policy, achieving sub-linear regret. On the contrary, when a cluster contains more states, the representative policy can fail to achieve optimal performance, if those are substantially different.

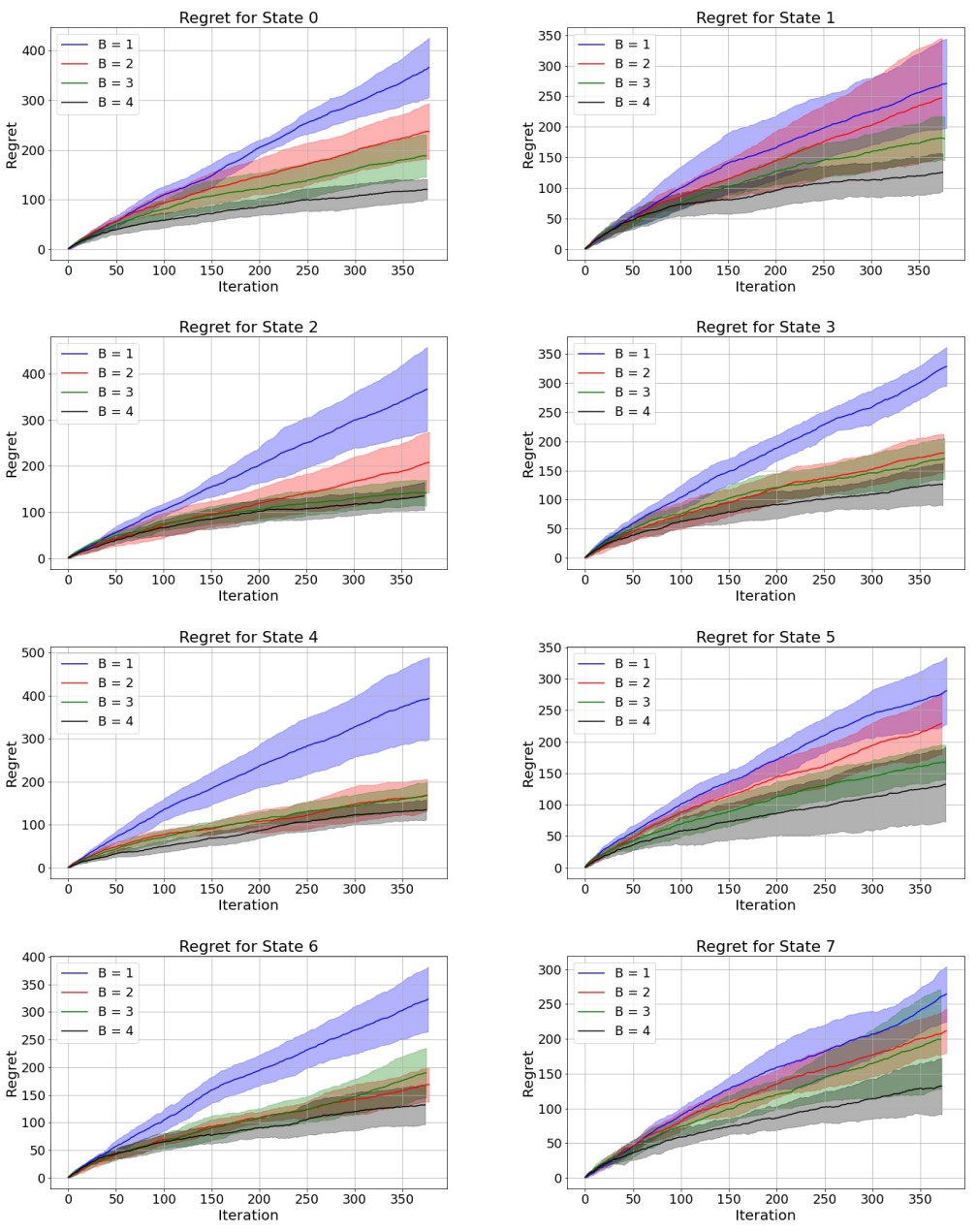

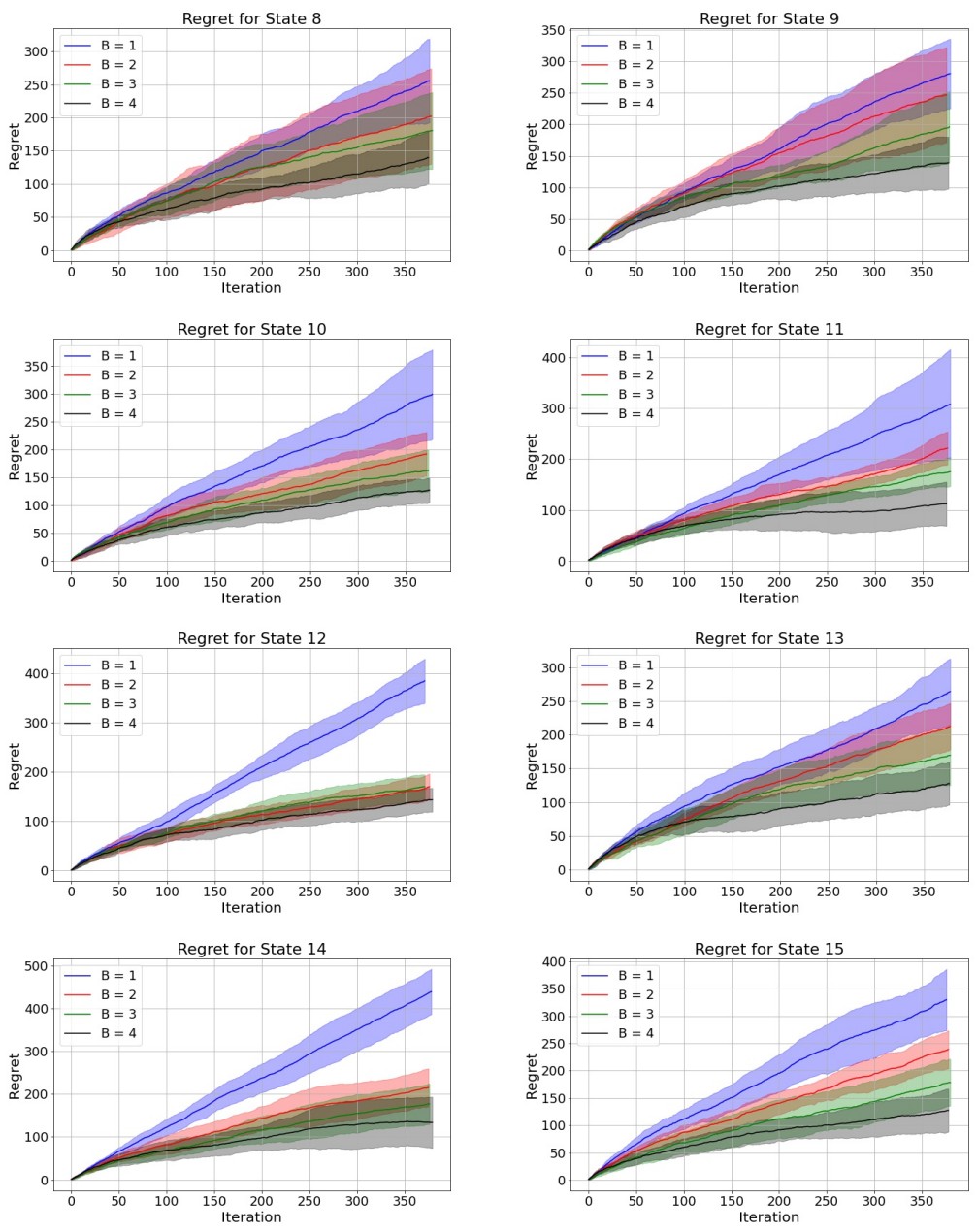

Figure 14: Deterministic experiment : cumulative average regret $\pm$ one std for cluster policy, when $B = \{1, 2, 3, 4\}$, from state $s = 0$ to state $s = 15$
.

# D  REGRET BOUND

To prove our statements, we define with $H_q(X)$ and $I_q(X;Y)$ the marginal entropy and mutual information w.r.t. to the joint probability $q$, i.e., $H(A^*) = H_{\pi^*}(A)$. We start by proving Lemma D.1.

**Lemma D.1.** *If the exact Thompson Sampling policies $\pi_t(a|s)$ achieve sub-linear Bayesian regret for all state $s \in \mathcal{S}$, then $\lim_{t \to \infty} I_{\pi_t}(S;A) = \lim_{t \to \infty} H_{\pi_t}(A) = H(A^*)$.*

*Proof.* First of all, we write $I_{\pi_t}(S;A) = H_{\pi_t}(A) - H_{\pi_t}(A|S)$. Following Theorem 2 from Kalkanli & Ozgur (2020), if $\pi_t(a|s)$ achieves sub-linear regret $\forall s \in \mathcal{S}$, then $\forall s \in \mathcal{S}$

$$\lim_{t \to \infty} \pi_t(a = a^*|s) = 1 \quad \text{when } a^* \text{ is the optimal arm}$$

$$\lim_{t \to \infty} \pi_t(a = a'|s) = 0 \quad \text{when } a' \neq a^*.$$

Consequently, we have that in the limit $\pi_t(a|s)$ is a deterministic function, thus

$$\lim_{t \to \infty} H_{\pi_t}(A^*|S) = 0,$$

which concludes our proof. ◻

Then, we prove Lemma 3.1, which is repeated below.

**Lemma 3.1** *If $R < H(A^*)$, then it is not possible to convey a policy $Q_t(s,a)$ that achieves sub-linear regret in all state $s \in \mathcal{S}$.*

*Proof.* Following Lemma 10 in Phan et al. (2019), if $\lim_{t \to \infty} \pi_t(a^*|s) = 1$ and $D_{KL}(Q_t||\pi_t) < \epsilon$, $\forall t$, for some $\epsilon > 0$, then $\lim_{t \to \infty} Q_t(a^*|s) = 1$, inducing $\lim_{t \to \infty} D_{KL}(Q_t||\pi_t) = 0$.

Now, for any policy $Q_t$, if the minimal rate to convey it, $\hat{R}_t = I_{Q_t}(S;A)$, is below the threshold of the optimal policy, i.e., $\hat{R}_t < H(A^*)$, by Eq. (2), we have

$$D_{KL}(Q_t||\pi^*) > 0, \tag{10}$$

which holds also in the limit. Otherwise, we would have a policy $Q_t$ s.t. $D_{KL}(Q_t||\pi^*) = 0$, which could be conveyed at a rate strictly below that characterized by Eq. (2), which would contradict with the definition of the rate-distortion function.

Consequently, by Lemma 10 in Phan et al. (2019) and Eq. (10), there cannot be any $\epsilon > 0$ s.t. $\lim_{t \to \infty} D_{KL}(Q_t||\pi_t) < \epsilon$; and hence, $\lim_{t \to \infty} D_{KL}(Q_t||\pi_t) = \infty$.

However, if $\lim_{t \to \infty} D_{KL}(Q_t||\pi_t) = \infty$, $\exists (s,a) \in \mathcal{S} \times \mathcal{A}$ s.t. $\lim_{t \to \infty} Q_t(a|s) = c > 0$, when $a \neq a^*$. This implies that, at step $t$, $Q_t(s,a)$ plays a sub-optimal arm in state $s$ with constant probability, and so it can not achieve sub-linear regret in all states. ◻

Finally, we provide a proof for Lemma 3.2, which we repeat below.

**Lemma 3.2** *If $R > H(A^*)$, then achieving sub-linear regret is possible in all states $s \in \mathcal{S}$.*

*Proof.* We define by $R^{\pi_t}$ the rate needed to convey the TS policy perfectly to the controller at time $t$, and let $\delta > 0$ s.t. $R = H(A^*) + \delta$, where $R$ is the available communication rate. We now provide a scheme that guarantees sub-linear regret.

First, generate $\rho_t$ parameters as in Sec. 3.3. As long as $R^{\pi_t} > R$, with probability $\rho_t$ play $a_t$ uniformly at random, and with probability $1 - \rho_t$, play according to a policy $Q_t(a|s)$, which satisfies the rate-distortion constraint $I(A;S) < R$ under $Q_t(a|s)$, which can be transmitted to the controller, and will have a bounded reverse KL divergence from the TS policy $\pi_t$ at time $t$. Following Lemma 14 in Russo (2016), in this way enough exploration is guaranteed for the TS policy to concentrate, and there exists a finite $t_0$ s.t. $\forall t > t_0$, $R^{\pi_t} < H(A^*) + \delta$. This means that, for the first $t_0$ rounds, both the target and the approximating policies are playing sub-optimal arms with non-zero probabilities. However, their average gap is within a constant, given the average rewards are bounded within $[0, 1]$. Then, $\forall t > t_0$ it is possible to play the exact TS policy, leading to the optimal policy for all future steps, and hence, a sub-linear regret. ◻

