# OpenReview forum: "Contextual Multi-Armed Bandit with Communication Constraints"
_ICLR.cc/2022/Conference — ICLR 2022 Submitted_

### Official Review · Reviewer_Cm2b · 2021-11-02

**Correctness:** 4
**Technical Novelty And Significance:** 3
**Empirical Novelty And Significance:** 3
**Recommendation:** 6
**Confidence:** 3

**Main Review:**

This is a well-written paper, and in particular I think the paper does a good job in the following aspects: 1. The RC-CMAB problem of interest is well-motivated in the introduction (e.g. with the personalized ad placement example); 2. The explanations for modelling the problem as a policy compression problem are clearly conveyed, and the relevant technical discussions on the distortion function as well as the clustering coding scheme (over states) are concise and easy to follow; 3. The analyses in the numerical result section are comprehensive and thorough, and I find the numerical illustrations related to the asymptotic rates to convey the considered policies (Fig. 4) quite insightful and interesting. Furthermore, I particularly liked the fact that the ideas behind the clustering coding scheme are simple and intuitive (since the idea of compressing states in clusters and transmitting a policy for each cluster is straightforward and natural), which is a huge plus from both a theory and practical perspective.

In terms of weaknesses, my main concern is that the paper lacks theoretical performance guarantees for the clustering coding scheme. I believe including a characterization of how regret depends on the number of states, the rate constraint, number of actions, etc. would provide us with a more in depth understanding of the performances of the clustering scheme in various scenarios. On a related note, I am specifically interested in how the proposed clustering schemes’ performance decays as the total number of states grows, because intuitively, more states would induce larger clusters and hence larger distortion in the policy defined over the compressed states (compared to the target policy). I am bringing this up because the paper uses personalized ad placements as a motivating example, and in such markets, typically the content owner’s valuation for a user view (i.e. impression) or conversion serves as the ``context’’ (or state), and may take a wide range of values. If providing theoretical analyses for the dependence on the state space is infeasible, at least having some experiments that address related issues would be beneficial.


**Summary Of The Paper:**

This work studies a rate-constrained contextual multi arm bandit (RC-CMAB) problem: the decision maker has to make action decisions for multiple parallel (independent and identical) CMAB problems (i.e. agents), but can only communicate the actions for each CMAB problem to a controller through a rate-constrained communication channel, from which the controller receives and decodes the actions of the decision maker, and applies such decoded decisions to each CMAB.  The paper first formulates the problem as a policy compression problem under an information theoretic framework, and then characterizes the optimal compression scheme for infinite agents. Next, the paper presents a practical coding scheme to communicate actions to finite agents under rate constraints, and finally benchmarks its performance against the compression scheme in the asymptotic regime.


**Summary Of The Review:**

The paper is well-written and well motivated. The technical discussions are concise and clear. The analyses for numerical results are comprehensive and thorough.

Weaknesses: the paper would be much stronger if it includes theoretical guarantees for the clustering scheme, and also results/discussions/experiments on how the cluster scheme’s performance decays with the number of states.

---

> ### Author Response · Authors · 2021-11-17
> **Response to the reviewer Cm2b**
>
> We thank the reviewer for the valuable comments and feedback. Our comments on the main concerns are available below.
>
> $\textbf{Comments}$
>
> As also pointed out in $\textit{Phan et al. (2019)}$, the theoretical characterization of regret as a function of the KL-Divergence between the optimal and the sampling policy is not easy to be derived, and it is something we’re actively investigating. Moreover, the clustering function does not provide guarantees in terms of achievable performance. However, to grasp the dependency of the performance w.r.t. to the number of per-agent bits, we carried out new experiments, which are reported in $\textbf{Appendix C.3}$. In particular, we remember that the key relation is between the number of per-agent bits and the mutual information between the state and the optimal action, as reported in the paper.  The experiment in $\textbf{Appendix C.3}$ considers a scenario with 16 states, 16 actions and 50 agents, still with a deterministic relation between state and action, as in the experiment in $\textbf{Sec. 4.3}$, and the number of bits used by the cluster policies varies from 1 to 4. As we can see from the plots, when the number of per-agent bits is low, i.e., 1 or 2, the regret is not sub-linear in many states. However, the policy with just 2 bits can achieve performance comparable with the 3 and 4 bits policies in some states, e.g., state 4 and 6. This is due to the fact that even in those cases with not enough bits, the posteriors still converges to good solutions, thus if a cluster contains, for example, one state, it will convey the best policy, achieving sub-linear regret. On the contrary, when a cluster contains more states, the representative policy can fail to achieve optimal performance, if those are substantially different.

---

### Official Review · Reviewer_vEGv · 2021-11-03

**Correctness:** 4
**Technical Novelty And Significance:** 3
**Empirical Novelty And Significance:** 2
**Recommendation:** 6
**Confidence:** 3

**Main Review:**

The paper is sound and of certain interest. One knock of this paper is that the results were largely adapted from known information-theoretic results: the optimal bound was derived based on Kramer&Savari; TS for RC-CMAB was based on the double minimization problem in Cover&Thomas as well as the well-known Blahut-Arimoto algorithm; the practical coding scheme was based on the well-known Lloyd algorithm for clustering. Putting these pieces together is not trivial, I must admit. However, the overall novelty is thus limited.

Another question is about the problem setting. The assumption that each agent's state $s_{t,i}$ is observed by the decision-maker, but not the agent itself, is very strange. Since the state is agent-dependent, agent $i$ generally already has access to the state $s_{t,i}$, which then makes this problem a decentralized MP-MAB problem (with contextual MAB). The authors should clarify why they make the assumption that only the central decision-maker can observe all agents' states, but not the agents themselves. In fact, the practical coding scheme is about sending the state (via a compact state representation) to the corresponding agent. This may not be necessary to begin with in practice. Also, with the aforementioned argument, the controller "decoder" function $g_t^{(N)}$ should have the joint states of all agents as an input. This would enhance the decoder design.

**Summary Of The Paper:**

This paper studies a CMAB problem where the actions for multiple agents are sent from the decision-maker over a rate-limited communication channel. The authors developed information-theoretic performance bound for Thompson sampling based policies, which reduce the problem to transmitting conditional probability distributions over a communication channel. A practical coding method was also developed. Numerical experiments were carried out to validate the proposed design.

**Summary Of The Review:**

A solid work overall, but with some questions on the novelty and assumptions. Will be open to author responses and adjust the score accordingly.

======

Post-rebuttal update: I thank the authors for addressing my concerns on the availability of state information to the decision-maker versus the agent. I hope that the authors add these details, especially the factory example, to the paper. My other reviews remain the same.

---

> ### Author Response · Authors · 2021-11-17
> **Response to the reviewer vEGv**
>
> We thank the reviewer for the detailed comments and valuable feedback. Here the response to the main concerns.
>
>
>
> **1) The paper is sound and of certain interest. One knock of this paper is that the results were largely adapted from known information-theoretic results: the optimal bound was derived based on Kramer&Savari; TS for RC-CMAB was based on the double minimization problem in Cover&Thomas as well as the well-known Blahut-Arimoto algorithm; the practical coding scheme was based on the well-known Lloyd algorithm for clustering. Putting these pieces together is not trivial, I must admit. However, the overall novelty is thus limited.**
>
>
> We believe we pose a novel problem with practical motivation, and provide a non-trivial solution to it; even though we do not employ a completely new technique for the solution. Our study is “complete”, in the sense that, we provide both information theoretic performance bounds, and practical algorithms for the problem at hand. Most importantly, we consider this as a first step towards studying a more general set of learning problems with communication constraints.
>
> **2) Another question is about the problem setting. The assumption that each agent's state $s_{t,i}$ is observed by the decision-maker, but not the agent itself, is very strange. Since the state is agent-dependent, agent $i$ generally already has access to the state $s_{t,i}$, which then makes this problem a decentralized MP-MAB problem (with contextual MAB). The authors should clarify why they make the assumption that only the central decision-maker can observe all agents' states, but not the agents themselves. In fact, the practical coding scheme is about sending the state (via a compact state representation) to the corresponding agent. This may not be necessary to begin with in practice.**
>
> The framework is motivated by those scenarios in which full context is only available at one point of the system, while the actors, i.e., the ones that can physically interact with the system, are connected through a communication channel. In the paper, we mentioned the ad recommendation example, which is a typical application of contextual multi-armed bandits, quoting:
>
> *“This scenario can model, for example, a personalized ad placement application, where the content owner observes the individual visitors to its website; and hence, has the context information, but must convey the ads that must be shown to each visitor to a separate entity that manages the marketing content.”*
>
> The framework can also model a remote control problem in a smart factory environment, where a centralized server-room has full knowledge of the underlying processes to be controlled, i.e., the state, and it has to inform a large number of actuators through a controller, i.e., a base station, about which actions they should take to monitor/control the underlying physical processes. The actuators in this setting may not have the sensing capability to observe the state, due to, for example, complexity constraints. The problem setting is also the one adopted in [1].
>
> Moreover, in the analyzed scenario, the decision-maker is connected to a single controller, as in the ad recommendation and in the smart factory systems, that receives the joint vector conveying the joint actions. Please note that the scheme is not transmitting the states to the agents, but the actions they should take. Indeed, the asymptotical solution is obtained by jointly encoding the policy related to all agents in a single vector, that is transmitted to the controller, that in turn decodes the whole message to extract the actions for all the agents.
>
> [1] T. Y.  Tung, S. Kobus, J. P. Roig & D. Gunduz..” Effective Communications: A Joint Learning and Communication Framework for Multi-Agent Reinforcement Learning over Noisy Channels.” IEEE Journal on Selected Areas in Communications, 39(8), 2590–2603. 2021.

---

> > ### Author Response · Authors · 2021-11-17
> > **Response to the reviewer vEGv (pt. 2)**
> >
> > **3) Also, with the aforementioned argument, the controller "decoder" function $g_t(N)$ should have the joint states of all agents as an input. This would enhance the decoder design.**
> >
> > We do agree that the proposed scenario, in which the states are fully observable by the agents, can be an interesting problem to be analyzed, but this is not the one considered in our paper. In that scenario, each agent can sample the corresponding action based on the observed state, leveraging its own policy. Admitting that a central authority can still have full knowledge of the rewards, the problem would be how to efficiently update the agents’ policies transmitting the effects of the sampled actions, so that they can update their policies accordingly. In this case, the compression scheme should exploit the estimated distributions of the rewards, given that for each state-action pair, there is an underlying unknown distribution governing the reward.
> >
> > However,  this is a different setting from ours, and a similar problem can be found in [2], in which the communication channel under consideration is from the agents to the central server, and the task is to compress the rewards feedback in order to still obtain good policy updates. We added a reference to clarify this difference in **Sec. 1.1 (Related Work)**.
> >
> > [2] O. A. Hanna, L. F. Yang and C. Fragouli. “Solving Multi-Arm Bandit Using a Few Bits of Communication.” Proceedings of the 38 th International Conference on Machine Learning, (ICML). 2021

---

### Official Review · Reviewer_mXFx · 2021-11-08

**Correctness:** 3
**Technical Novelty And Significance:** 3
**Empirical Novelty And Significance:** 3
**Recommendation:** 6
**Confidence:** 2

**Main Review:**

Strength:
* The paper proposes an original problem that combines the challenges of bandit modelling and information constrained communication problems. I particularly like that this problem has achievability constraints and elegantly makes use of information theory to solve a contextual bandit problem.
* The problem is motivated by a real real-world task

Weaknesses:
* There are some inconsistencies and imprecisions in the writing that make it hard to follow. Section 2 defines a lot of quantities and regrets and gains, etc. that are not really analyzed afterwards, or I missed it: what do you mean to say about G and \rho and R ? Also, how is R defined exactly (the “communication rate”), is it some ratio of N and B or something ?
* The paper does not have regret bounds, I guess it was out of your scope, but can you comment on that ? You briefly talk about problems that are “achievable” but then we don’t really know what kind of regret we could expect it seems.
* Algorithm needs to be clearly outlined: You give very detailed explanations on your algorithms but for the sake of readability we need a pseudo-code. At the moment, I am struggling mending the pieces between all the different optimization procedures to solve and what needs to be done when and by whom. Can you please outline a pseudo-code and upload it so I can check that my understanding corresponds to what you are doing ?
* Minor: Lower-level literature review: I feel like the background is given in the related work but I am surprised you were not able to connect your work better with the existing literature on contextual bandit. Perhaps contextual bandits with context uncertainties, or Partially Observable MDPs ?


Minor:
* section 3.3 : “posterior \pi convergences” -> “converges”

**Summary Of The Paper:**

This paper studies a contextual bandit problem where the decision-maker must communicate its intended actions (given observations of the contexts) to a controller through a constrained communication channel. The original part of the paper is that the “bandit algorithm” must encode its actions into a compressed version that then serves to the controller.



**Summary Of The Review:**

TL;DR: The paper is a bit hard to follow for the non-expert in compression that I am (arguably like many of the bandit community) so I am not sure of my evaluation and I hope to be able to clarify it during the rebuttal / discussion phase. I cannot make a strict recommendation for now but I will update my score after the rebuttal phase.

---

> ### Author Response · Authors · 2021-11-19
> **Response to the reviewer mXFx**
>
> We want to thank the reviewer for the detailed feedback and analysis. Here below the reponse to the reported concerns.
>
> **Major**
>
> **1) There are some inconsistencies and imprecisions in the writing that make it hard to follow. Section 2 defines a lot of quantities and regrets and gains, etc. that are not really analyzed afterwards, or I missed it: what do you mean to say about $G$ and $\rho$ and $R$ ?**
>
> The quantities defined in **Sec. 2**, the rate $R$ and the regret $\rho$, are used to identify the trade-off between the available communication rate, and the achievable regret. Our goal is to study the trade-off between these two extreme points, and in the revised version, we have identified two distinct rate regions resulting in linear and sub-linear regret behaviour, respectively.
>
> We added a clarification in **Sec. 3.4**, together with the asymptotic regret analysis.
>
> **-Also, how is $R$ defined exactly (the “communication rate”), is it some ratio of $N$ and $B$ or something ?**
>
> Rate is defined as the number of bits per agent that can be transmitted from the decision-maker to the controller, i.e., for a rate of $R$, the total number of different messages $B$ that can be conveyed is limited by $2^{nR}$, that is explicitly defined at the end of **Sec. 2.2.**
>
>
> **2) The paper does not have regret bounds, I guess it was out of your scope, but can you comment on that ? You briefly talk about problems that are “achievable” but then we don’t really know what kind of regret we could expect it seems.**
>
> Please see the general comment *Response to all Reviewers*.
>
> **3) Algorithm needs to be clearly outlined: You give very detailed explanations on your algorithms but for the sake of readability we need a pseudo-code. At the moment, I am struggling mending the pieces between all the different optimization procedures to solve and what needs to be done when and by whom. Can you please outline a pseudo-code and upload it so I can check that my understanding corresponds to what you are doing ?**
>
> We thank you for pointing it out. We added the pseudocodes in **Appendix B.1**, and **B.2.**
>
> **Minor**
>
> **1) Lower-level literature review: I feel like the background is given in the related work but I am surprised you were not able to connect your work better with the existing literature on contextual bandit. Perhaps contextual bandits with context uncertainties, or Partially Observable MDPs ?**
>
> We extended **Sec. 1.1** (Related Work) to better connect our research with other multi-agent and contextual bandits papers.
>
> **2) Section 3.3 : “posterior \pi convergences” -> “converges”**
>
> We thank you for spotting out the typo, we fixed it.

---

> > ### Comment · Reviewer_mXFx · 2021-11-19
> > **Thank you for your precisions, I'll have a second look at the paper to reevaluate my score**
> >
> > Thank you for your answers to my questions. I am sorry for misunderstanding the definition of R. Now, I see that the rate R is simply a number that defines an upper bound on B, which is the important quantity quantifying the number of encoded messages. I still feel that could be made a bit clearer, for instance it seems to mean that B=\lfloor 2^{nR} \rfloor, is that right ? Or is there a reason B would be chosen smaller than the limit ?
> >
> > Also, regarding the pseudo-codes, I must admit I was hoping that the TS step would be made a bit clearer because it does not seem obvious to me what is the posterior of each action given that a clustering step is also performed.
> > On a second thought, and now that I understand better the process, it seems to me that this method is really close to "Online clustering of bandits" (https://arxiv.org/abs/1401.8257) by Gentile et al 2014. They have a UCB type of approach but otherwise they also cluster contexts and decide on actions based on the clustering. I don't know if a TS version of that paper exists but if it does, it would be very related to yours, I'll do a quick research.
> >
> > I'll have a second look at your paper as well to reevaluate my score.
> > Thank you,
> > Reviewer

---

> > > ### Author Response · Authors · 2021-11-19
> > > **Response to the Reviewer mXFm**
> > >
> > > We comment on the last concerns below.
> > >
> > > **1)Now, I see that the rate $R$ is simply a number that defines an upper bound on B, which is the important quantity quantifying the number of encoded messages. I still feel that could be made a bit clearer, for instance it seems to mean that $B=\lfloor 2^{nR} \rfloor$, is that right ? Or is there a reason $B$ would be chosen smaller than the limit ?**
> > >
> > > Yes, that’s correct. For a given rate R, the maximum B is given by $B=\lfloor 2^{nR} \rfloor$, and in general there is no gain in choosing a smaller B. Our definition follows the standard definition of achievability (see, for example, [Ref]), with average regret replacing a more standard distortion function in rate-distortion theory:
> > >
> > > [Ref] Thomas M. Cover and Joy A. Thomas. 2006. Elements of Information Theory (Wiley Series in Telecommunications and Signal Processing). Wiley-Interscience, USA.
> > >
> > > **2)Also, regarding the pseudo-codes, I must admit I was hoping that the TS step would be made a bit clearer because it does not seem obvious to me what is the posterior of each action given that a clustering step is also performed.**
> > >
> > > The practical coding algorithm is the one explained in **Algorithm 2**, **Appendix B.2.** As we can see, at the end of each iteration, the decision-maker updates the posterior $\pi_t$ using classical TS, as explained in **Sec. 3.1.** However, the actors cannot adopt that policy, given the rate constraint, and so the decision-maker computes the indexes of the clustered policies to be transmitted, in line
> > >
> > >  > **$\forall s_{t,i}$ compute the index $j(s_{t,i})$ indicating its belonging cluster**
> > >
> > >
> > > and transmits them to the controller over the channel.
> > >
> > > Now, the cluster policies are constructed using the updated $\pi_t$ as reference, following the steps in **Sec. 3.5.** However, this construction is not made at each time step, but only when the real policy $\pi_t$ has changed substantially, w.r.t. to the last one used to construct the clusters, as explained in the *if statement* in **Algorithm 2.**
> > >
> > > **3)On a second thought, and now that I understand better the process, it seems to me that this method is really close to "Online clustering of bandits" (https://arxiv.org/abs/1401.8257) by Gentile et al 2014. They have a UCB type of approach but otherwise they also cluster contexts and decide on actions based on the clustering. I don't know if a TS version of that paper exists but if it does, it would be very related to yours, I'll do a quick research.**
> > >
> > > We thank you for the reference. Even though the scenario analyzed in the paper is very interesting, and somehow related to ours, we think that there are some fundamental differences:
> > >
> > > 1. They focus on linear bandits, with a parametric expression of the contexts, whereas in our framework states are taken from a finite set, and a different TS policy is followed at each state.
> > >
> > > 2. They assume there is an inherent structure among the users, quoting *"We assume the user behavior similarity is encoded as an unknown clustering of the users"* which is not the case in our model, where the clustering is performed by the decision-maker, and the number of clusters is imposed by the rate constraint, not by the underlying bandits game.
> > >
> > > 3. Mainly, they assume an underlying clustering scheme in the context space, imposed by the problem. Instead, our coding scheme clusters the states based on action policies (**Sec. 3.5**), which are k-dimensional vectors in the k-simplex, and are induced by TS at the decision-maker, as explained in **Sec. 3.1.** Consequently, our scheme employs a clustering scheme which is based on the stochasticity of the TS sampling strategy, and uses the KL-divergence as the similarity measure in the clustering space.

---

### Official Review · Reviewer_CLeX · 2021-11-09

**Correctness:** 4
**Technical Novelty And Significance:** 3
**Empirical Novelty And Significance:** 2
**Recommendation:** 5
**Confidence:** 4

**Main Review:**

Major comments:
1. Most importantly, there is no complete theoretical analysis (e.g. the upper bound/lower bound on the regret of the algorithm.)
1. The algorithm should be explicitly written out with its pseudocode.
1. By the end of Section 1, it states that the communication is one-way from the decision-maker towards to controller. However, as the decision-maker can observe the realization history, it is a bit strange not to consider that as communication. Besides, how the goal of this work is different from existing papers is confusing.
2. The problem setup is not clear: is the state i.i.d. sampled from the distribution $P_S$ or not?
3. The relation between the constraint quantities $(\rho,R)$ and the discussions in Section 3 is not clear.
4. The constraints seem to be placed in the posterior probabilities involved in the TS algorithms. What if we consider UCB-class algorithms?
5. The definitions of $D_\alpha$ and reverse KL divergence $D_{KL}$ is missing. Usually, we denote the original KL divergence by $D_{KL}$.
6. In Section 3.3, it states ''If $S$ and $A$ are independent...'', I don't think it's a common assumption in contextual bandits.

Minor comment:
1. The citation style is not so good: for instance, it should be "In Phan et al. (2019) ..." instead of "In (Phan et al., 2019)..."


**Summary Of The Paper:**

This paper focus on the contextual multi-armed bandit with communication constraints. It provides some discussions and numerical results.

**Summary Of The Review:**

Overall, this works lacks complete analysis, which is not usual in a bandit paper, and the relation between the background and the discussions is not clear. Hence, the contribution of the work is not clear. The detailed comments are as above.



================================
I appreciate the efforts of authors to add on theoretical results during the rebuttal period, and decide to raise the ranking from 3 to 5.

---

> ### Author Response · Authors · 2021-11-19
> **Response to the reviewer CLeX**
>
> **1 ) Most importantly, there is no complete theoretical analysis (e.g. the upper bound/lower bound on the regret of the algorithm.)**
>
> Please see the general comment *Response to all Reviewers*.
>
> **2) The algorithm should be explicitly written out with its pseudocode.**
>
>  We added the pseudocodes in **Appendix B.1**, and **B.2**.
>
> **3) By the end of Section 1, it states that the communication is one-way from the decision-maker towards to controller. However, as the decision-maker can observe the realization history, it is a bit strange not to consider that as communication. Besides, how the goal of this work is different from existing papers is confusing.**
>
> It is correct that the information flow is two-ways, but we were referring explicitly to the “active” communication channel, where the information needs to be encoded; particularly comparing our model with the recent literature on multi-agent RL problems with two-way communication between agents that enables better coordination. We have updated the sentence as *“.. we focus on a particular setting, in which the communication channel is one-way, from the decision-maker to the controller…”*.
>
> **4) The problem setup is not clear: is the state i.i.d. sampled from the distribution PS or not?**
>
> We thank you for pointing it out. Yes, the states are sampled iid and we added this missing point in **Sec. 2.1.**
>
> **5) The relation between the constraint quantities (ρ,R) and the discussions in Section 3 is not clear.**
>
> In general there is an inherent trade-off between the available communication rate $R$ and the achievable regret $\rho$. It is clear that, when the communication rate is zero, the controller will simply be limited to random actions, and the regret will grow linearly according to the average reward. On the other hand, when the communication rate is sufficient (i.e., $R \geq \log K$), then the scenario reduces to the setting in which the decision-maker and the controller are colocated, and a sublinear regret  can be achieved. Our goal is to study the trade-off between these two extreme points, and in the revised version, we have identified two distinct rate regions resulting in linear and sub-linear regret behaviour, respectively.
>
> We clarified this connection in **Sec. 3.4**.
>
>
> **6) The constraints seem to be placed in the posterior probabilities involved in the TS algorithms. What if we consider UCB-class algorithms?**
>
> The main reason why we adopted a TS-based algorithm is that its exploration strategy, driven by the sampling process, induces a distribution over the actions, which lends itself naturally to an information theoretic analysis when considered under a rate constraint. Even though a heuristic distribution may be constructed using a UCB-like algorithm considering the action scores, they may not perform competitively, whereas TS is a well-known and competitive algorithm for MAB problems. For further comparison of UCB and TS algorithms, and the advantages of the latter, please refer to
>
> *D. Russo, B. Van Roy (2014) Learning to Optimize via Posterior Sampling. Mathematics of Operations Research 39(4):1221-1243.*
>
> **7) The definitions of $D_{\alpha}$ and reverse KL divergence $D_{KL}$ is missing. Usually, we denote the original KL divergence by $D_{KL}$.**
>
> We thank you for warning about this missing point. We added and clarified the divergence definitions in **Sec. 3.3.**
>
> **8) In Section 3.3, it states ''If S  and A are independent...'', I don't think it's a common assumption in contextual bandits.**
>
> The independence assumption in **Sec. 3.3** is used just to introduce an edge case, in order to explain the framework. The independence between S and A **is not an assumption of our model or method.**

---

> > ### Author Response · Authors · 2021-11-19
> > **Response to the reviewer CLeX (pt. 2)**
> >
> > **Minor**
> >
> > **1) The citation style is not so good: for instance, it should be "In Phan et al. (2019) ..." instead of "In (Phan et al., 2019)..."**
> >
> > Thank you, we fixed the citation format.
> >
> > **We want to the thank the reviewer for the careful analysis and valuable insights, which made us improve our work.**

---

### Author Response · Authors · 2021-11-19
**Response to all Reviewers**

**- In light of your valuable comments and feedback, we performed an extensive analysis on the achievability of sub-linear regret with a rate-constrained one-way communication channel from the decision-maker to the controller.**

In particular, we provide a value for the minimum rate, i.e., $H(A^*)$ - that is the entropy of the marginal distribution (over the state) of the optimal action - needed to obtain sub-linear regret. This means that, if the available communication rate $R$ is s.t. $R<H(A^*)$, a sub-linear regret is not possible ( **Lemma 3.1**).

Moreover, we show its counterpart; that is, for all rates $R> H(A^*)$, it is possible to achieve sub-linear regret (Lemma 3.2).

We believe that this analysis adds a valuable theoretical component to our work, and hope that satisfies reviewers’ expectation regarding the lack of regret analysis. The proofs of the Lemmas can be found in **Appendix D**.

**- Other changes to the paper**

We refer to the new theoretical results also in the abstract, as well as in the conclusion. Moreover, the state representation experiment, that was **Sec. 4.1** in the old version, has been moved to **Appendix C.1**. Consequently, the new experiments on *Regret vs Number of classes*, that was previously contained in Appendix C.3, is now in Appendix C.4. The sub-sections of **Sec. 4** have been also renamed accordingly. In end, we added the pseudocodes for both the asymptocally-optimal and cluster policies in **Appendix B.**

---

### Decision · Program_Chairs · 2022-01-20

**Decision:**

Reject

**Comment:**

Summary: This paper studies a contextual bandit problem where the decision-maker must communicate its intended actions (given observations of the contexts) to a controller through a constrained communication channel. The original part of the paper is that the “bandit algorithm” must encode its actions into a compressed version that then serves to the controller. In that sense, the controller must cluster the problems for the decision maker to simplify communication.

Discussion: Most reviewers appreciated that the paper is well-written and proposes an original problem. The main commonly issue is that of a lack of regret analysis. The authors included an additional theoretical result giving a necessary condition for sublinear regret but the committee would still appreciate a more in-depth study of the performance of the proposed algorithm, given that the condition is satisfied. One possible direction is to connect this work with the literature on "Clustering of bandits" (CoB) as raised by reviewer mXFx. The authors claim that this paper is only mildly related but the committee would kindly insist that linear rewards are just a generalization of multi-armed bandits, that there is also a finite state space in CoB (finite population) and it seems possible to reduce the proposed problem to CoB under some assumptions. In that regards, it would be important that a more thorough review and comparison of that line of work is done in the main paper (note also "Latent Bandits" and related papers), even though we agree that the proposed approach is different and we appreciate its originality.

Overall, the paper is borderline, and the committee did not reach a consensus.

Decision: Reject